# Extreme conditions affect neuronal oscillations of cerebral cortices in humans in the China Space Station and on Earth

Peng Zhang[1,2], Juan Yan[3], Zhongqi Liu[1,2], Hongqiang Yu[4], Rui Zhao[4] & Qianxiang Zhou [1,2✉]

Rhythmical oscillations of neural populations can reflect working memory performance. However, whether neuronal oscillations of the cerebral cortex change in extreme environments, especially in a space station, remains unclear. Here, we recorded electroencephalography (EEG) signals when volunteers and astronauts were executing a memory task in extreme working conditions. Our experiments showed that two extreme conditions affect neuronal oscillations of the cerebral cortex and manifest in different ways. Lengthy periods of mental work impairs the gating mechanism formed by theta-gamma phase-amplitude coupling of two cortical areas, and sleep deprivation disrupts synaptic homeostasis, as reflected by the substantial increase in theta wave activity in the cortical frontal-central area. In addition, we excluded the possibility that nutritional supply or psychological situations caused decoupled theta-gamma phase-amplitude coupling or an imbalance in theta wave activity increase. Therefore, we speculate that the decoupled theta-gamma phase-amplitude coupling detected in astronauts results from their lengthy periods of mental work in the China Space Station. Furthermore, comparing preflight and inflight experiments, we find that long-term spaceflight and other hazards in the space station could worsen this decoupling evolution. This particular neuronal oscillation mechanism in the cerebral cortex could guide countermeasures for the inadaptability of humans working in spaceflight.

[1] School of Biological Science and Medical Engineering, Beihang University, Beijing 100191, China. [2] Beijing Advanced Innovation Center for Biomedical Engineering, Beihang University, Beijing 100191, China. [3] China CDC Key Laboratory of Radiological Protection and Nuclear Emergency, National Institute for Radiological Protection, Chinese Center for Disease Control and Prevention, Beijing 100088, China. [4] China Astronaut Research and Training Center, Beijing 100193, China. ✉email: zqx_buaa@163.com

Working memory (WM) performance, which describes the human capacity to prospectively store information and translate it into an appropriate behavioral response, can be reflected in rhythmical oscillations of neural populations[1]. However, whether neuronal oscillations of the cerebral cortex during WM tasks change in extreme environments[2–7], especially in a space station, remains unclear. First, most missions executed in a space station, with insufficient personnel and a high workload, require effort over an extended period. Nevertheless, the willingness to work is not static due to the prolonged time of the mission[8]. Individuals intend to give up on a task or take a break even though the objective difficulty remains the same[9,10]. Current electrophysiological accounts propose that persistent neuronal firing is the basic process underlying the temporary storage of information in a widespread neural network[1]. Thus, sustained neuronal firing is a WM hallmark when the information is maintained during the delay before a response[11]. Single-neuron recordings in humans have indicated that concept neurons with high stimulus selectivity were persistently active during WM maintenance[12,13], and no workload dependence was noted[14,15]. In our ground experiment for volunteers, the mental task was extended to 90 min, accompanied by acquisitions of electroencephalography (EEG) signals, to investigate whether neuronal oscillations of the cerebral cortex during WM depend on the length of work.

Second, sleep deficiency[16] is prevalent during mission implementation on the International Space Station. Although the International Space Station provides light-attenuated and sound-attenuated sleep conditions for astronauts, sleep deprivation and fatigue remain common complaints among astronauts[17]. Falling asleep does not mean the brain switches off; instead, the neural circuitry is actively working through a complex architecture of sleep stages, which includes a series of carefully interleaved cognitive and physiological processing states[18]. Lack of sleep impairs many aspects of cognition[19–24], including WM performance, because sleep promotes memory integration and consolidation, gist extraction, and the ability to disregard distractions. Recent studies have shown that synaptic junctions in the cortex are also dependent on sleep[25,26] and that experimental sleep-wake disruption causally modulates β-amyloid dynamics[27]. Furthermore, the association between β-amyloid pathology of the medial prefrontal cortex and impaired memory consolidation is not direct but instead statistically dependent on the intermediary factor: the diminished slow-wave activity in nonrapid eye movement sleep[28]. In another ground experiment, the awake period of healthy volunteers was prolonged to 36 h, which was expected to elicit abnormal cortical activity in a neuronal oscillation mechanism.

Last, the astronauts' nutritional supply[29–31] and psychological situations[32–35] may also negatively affect WM performance. In research on nutritional supply[36–38], experimental data suggest that hippocampal synaptic plasticity plays a critical role in acquiring new information on memory[39], while it is also associated with nutritious or metabolic disorder signals[40,41]. Notably, glucose regulates brain function, especially memory, and astrocytic lactate production is also essential for memory[42]. Biomolecules[43] inside the brain or cortex might particularly account for the association between observed neuronal oscillatory and cognitive changes[44]. Regarding research on psychological situations, space psychology, a focused frontier field in human sciences, was established with the launch of Yuri Gagarin on April 12, 1961[45], and living in a confined and isolated environment impacts mood and cognition[4]. Relevantly, the brain balance between emotions and cognition is disrupted in emotional disorders, such as anxiety and depression, and the nervous system is biased toward processing negative psychological information[46]. Therefore, healthy volunteers were recruited for complete fasting

for up to 7 days in our last ground experiment, and abnormal mood samples from all ground experiments were used to analyze the neuronal oscillations in the cerebral cortex during WM tasks in extreme environments.

Figure 1 presents a schematic workflow for this study. The experimental results showed that a lengthy mental task and sleep deprivation affect neuronal oscillations of the cerebral cortex in different ways. First, theta phases reflected in the frontal-central cortex (FCC) surface regularly modulate gamma amplitudes reflected in the temporal-parietal cortex (TPC) surface, while this mechanism was gradually decoupled in the lengthy mental task experiment. Second, theta wave activity (TWA) reflected in the FCC surface increased within the optimal window limits during WM, and sleep deprivation disrupted this homeostasis, leading to an substantial increase in TWA. Last, participants' data in the complete fasting experiment and abnormal mood samples from all ground experiments showed that the decoupling of theta-gamma phase-amplitude coupling (PAC) and the change in TWA were independent of the nutritional supply or psychological situation. These findings demonstrate that human mental work at the China Space Station is a primary factor influencing neuronal oscillations, reflected in the PAC temporal dynamics between the FCC and TPC cortical surfaces.

## Results

### A lengthy mental task decoupled the theta-gamma PAC reflected between the frontal-central and temporal-parietal cortical surfaces

Given that one gating mechanism of prefrontal cognitive resources that is useful for remote neocortical areas in WM tasks is controlled by cross-structure oscillatory coupling or decoupling between frontal theta phase and temporal gamma amplitude[47–49], we set out to monitor whether this kind of PAC was independent of working length. To this aim, we measured the scalp EEG of 54 healthy volunteers performing two-back tasks. In addition, the lengthy mental task was embedded in two-back tasks to form an extended working condition (Fig. 1a). After the first two-back task (I-2B task), the lengthy mental task was performed prior to the second two-back task (II-2B task). The mean accuracy rate (Supplementary Fig. 1a) was generally higher for the I-2B task $(85.25\% \pm 2.78\%)$ than for the II-2B task $(80.75\% \pm 4.18\%)$, as indicated by analysis of variance (ANOVA) $(F_{1/53} = 47.20, p < .0001, \text{Fig. 2a})$. These results indicate that lengthy mental tasks hinder the memory-related behavioral performance of participants during two-back tasks.

To assess whether PAC detected across cortical surfaces, we employed a cross-frequency measure[49] for analyzing the coupling modulation in every trial (see methods section for details). This study focuses primarily on PAC of neuronal oscillations between low-frequency (theta, 6 Hz) and high-frequency range (gamma, 30–80 Hz), an indicator of increased neuronal firing and information processing[50], although other types of coupling across frequencies exist[51,52] (see supplementary materials for results). The chord diagram (Fig. 2b) of the I-2B task from the ergodic analysis showed TPC electrode clusters experienced a great majority of gamma-amplitude modulation by the theta phase obtained from the FCC. For each participant (Fig. 2d), the number of coupled electrodes, when the corrected p-value of Wilcoxon tests was <0.05, was significantly reduced compared to that before the lengthy mental task. For all 54 participants, during the retention interval in the I-2B task, a burst of TPC gamma activity locked to the negative trough (Fig. 2b, f) of the FCC theta band compared to shifted data (for all electrode permutations: most $|Z| > 1.97$ and $p < .05$, FDR-corrected, Supplementary Fig. 1b). However, during the II-2B task, this cross-structure oscillation mechanism was decoupled (Fig. 2c, g) after the lengthy

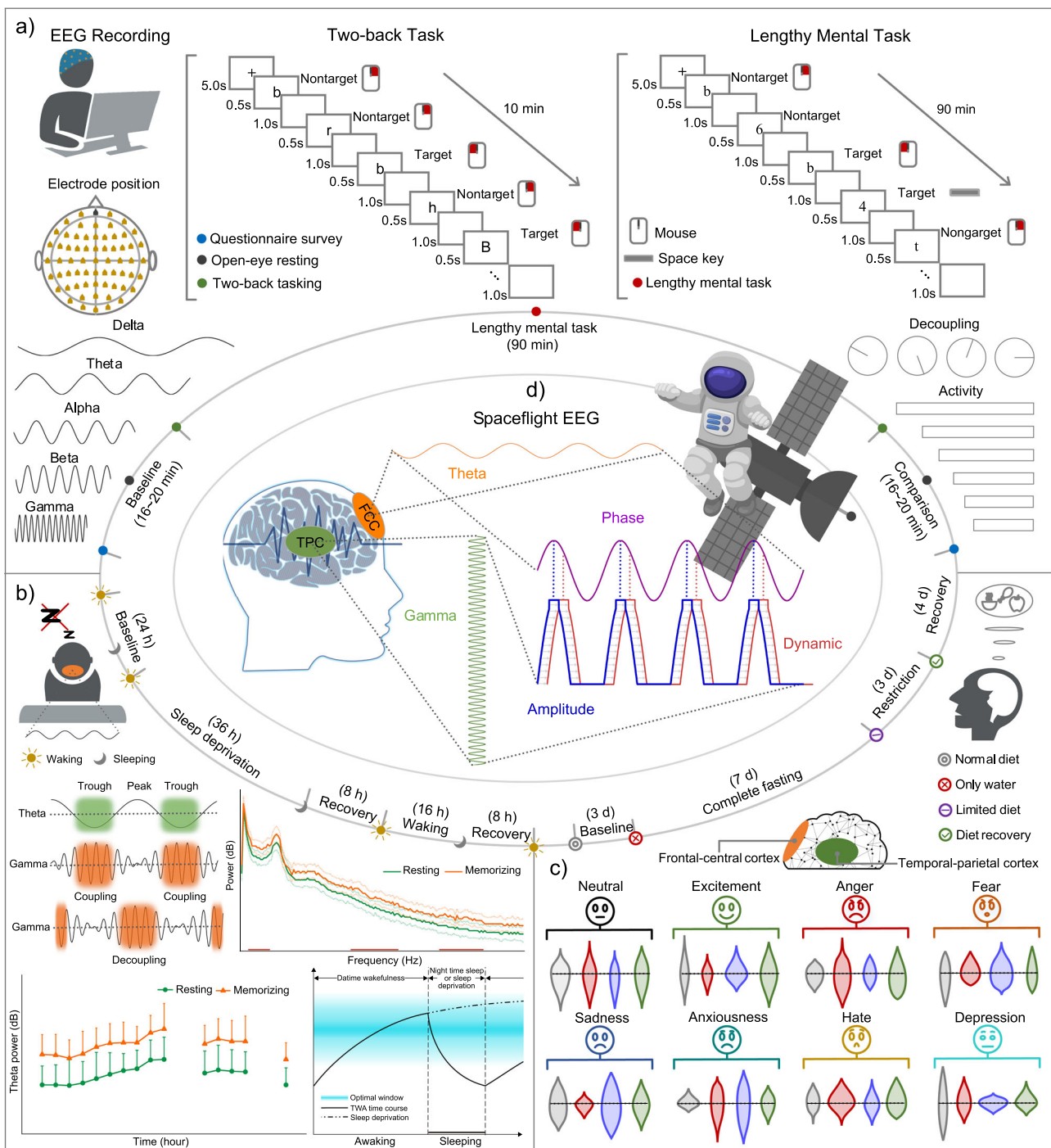

**Fig. 1 Experiments and task design. a** Before and after all experiments described below, EEG was simultaneous recording when participants performed the two-back task of letters. Participants were asked to press the left mouse button using the right index finger when the current letter matched the letter presented in two intervals (3 s) previously. Otherwise, press the right mouse button using the right middle finger. Participants performed the lengthy mental task in the first experiment, accompanied by acquisitions of EEG. Participants were asked to press the left mouse button using the right index finger when the current letter matched the letter presented in two intervals (3 s) previously. Otherwise, press the right mouse btton using the right middle finger. In addition, if the parity of the current numeral matched that of the numeral presented in two intervals (3 s) previously, participants were asked to press the keyboard's space bar. Otherwise, no reaction was required. EEG was used in analyses of the theta-gamma phase-amplitude coupling (PAC) and theta wave activity (TWA). **b** Participants experienced 36 h sleep deprivation in the second experiment, and **c** participants experienced complete fasting of 7 days in the third experiment, and EEG was still used in analyses of the PAC and TWA. **d** Theta-gamma PAC during the two-back task was decoupled when astronauts worked in the China Space Station for recent Shenzhou missions.

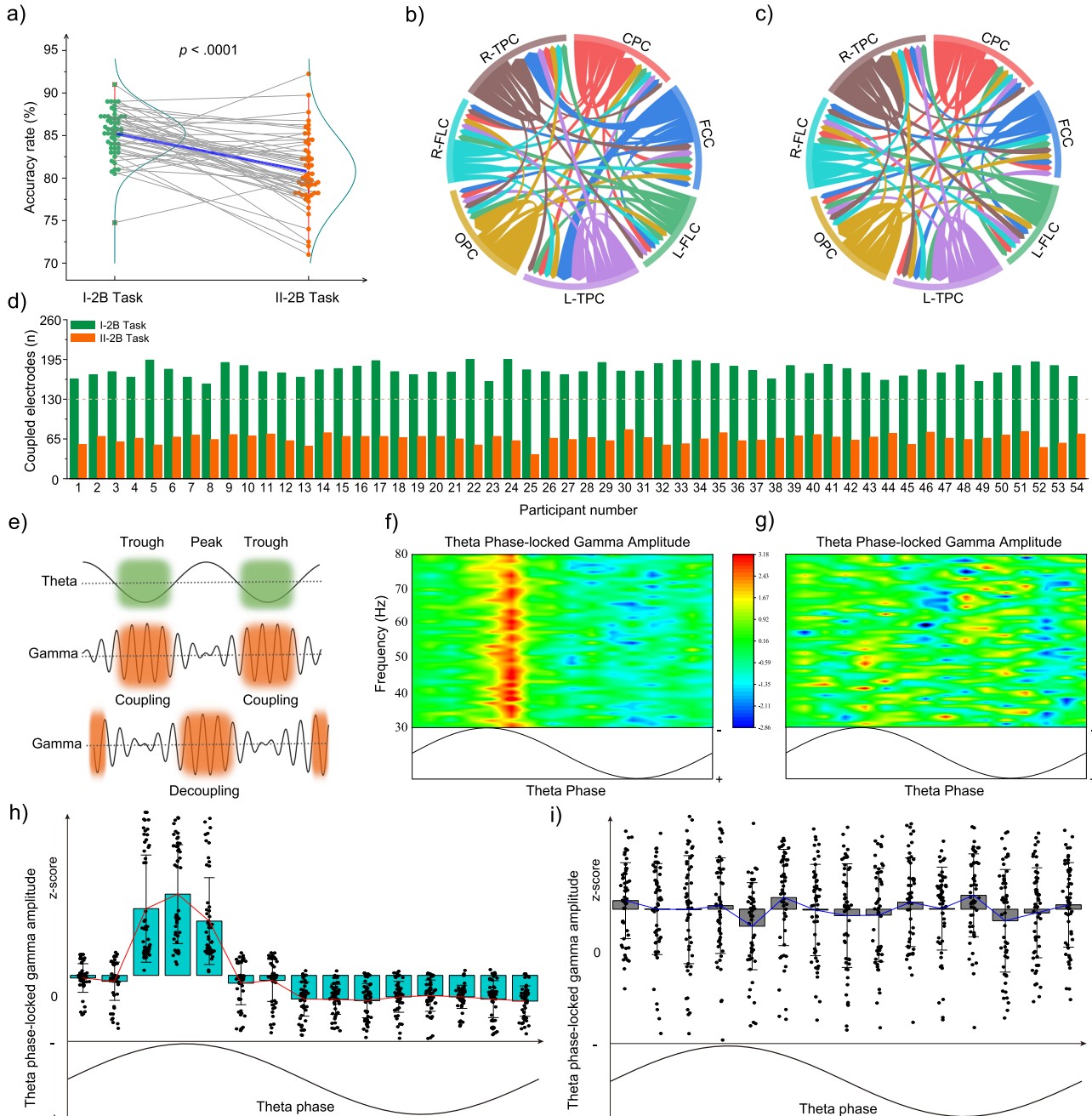

**Fig. 2 Results of the lengthy mental task.** $n = 54$, **a** after the lengthy mental task, behavior performance, the mean accuracy rate of the two-back task, was significantly declined indicated by a one-way repeated ANOVA. I-2B and II-2B tasks represented the two-back tasks before and after the lengthy mental task, respectively. **b** The chord diagram of the I-2B task from the ergodic analysis shows electrode clusters in the temporal-parietal cortex (TPC) surface having a great majority of gamma-amplitude modulation by theta phase obtained from electrode clusters in the frontal-central cortex (FCC) surface. **c** During the II-2B task, this cross-structure phase-amplitude coupling (PAC) was decoupled after the lengthy mental, indicated by the change in the thickness of the connecting lines. **d** The number of coupled electrodes significantly reduced for each participant compared to before the lengthy mental task. Green represented the I-2B task, and orange represented the II-2B task. **e** A challenging working memory such as the mental manipulation of a two-back task will require maximal allocation of cognitive resources, which is achieved by alignment of frontal-central and temporal-parietal neural firing, enabled by nesting TPC gamma activity into the excitatory trough of FCC theta phase. However, this coupled phase position removes from the trough after the length mental task, and it is improbable that much neural communication between the frontal-central and temporal-parietal cortex occurs in this condition. **f** Instantaneous amplitude values for frequencies >30 Hz from a TPC electrode were locked to the negative peak of a theta cycle from an FCC electrode. **g** This phenomenon disappeared after the lengthy mental task. **h** Cyan bars (means ± standard deviations) further indicated that the theta phase in FCC surface significantly modulated gamma amplitude in terms of the average state in TPC surface for the I-2B task. **i** This phenomenon disappeared in the II-2B task, indicated by grey bars (means ± standard deviations).

mental task (for all electrode permutations: most $|Z| < 1.97$ and $p > .05$, FDR-corrected, Supplementary Fig. 1b). The theta-gamma PAC also existed within the same temporal-parietal, central-parietal, or occipital-parietal cortex structure, but this coupling was independent of working length (Fig. 2b, c). To further evaluate whether gamma amplitude is significantly modulated by the theta phase, one-way repeated-measures ANOVAs with the factor PHASE BIN (segment 1-15: each bin covering 24°) with frontal-central theta phase-sorted gamma amplitude averaged within temporal-parietal electrodes as the dependent variable were performed. This result further indicated that the theta phase of the FCC significantly modulated gamma amplitude in terms of the average state of TPC for the I-2B task (ANOVA: $F_{14/742} = 94.06$ and $p < 0.0001$, Fig. 2h) rather than the II-2B task (ANOVA: $F_{14/742} = 1.69$ and $p = 0.0673$, Fig. 2i).

**Lengthy mental task performance showed dynamic decoupling of theta-gamma PAC reflected between the FCC and the TPC.** Participants performed the lengthy mental task continuously for 90 min; this time period was successively divided into nine blocks (TIME: time 1~time 9), to induce a behavioral or cognitive variation with the extended time of mental memory work. A letter or numeral was alternately shown, and participants were instructed to determine if a current letter was the same as the prior one (CONDITION: heavier load) or the parity of a current number was the same as the prior one (CONDITION: easier load), which produced a condition variation in the cognitive task (Fig. 1a). Theta phases of 13 frontal-central electrodes (Fz, F1-4, FCz, FC1-4, AFz, and AF3-4) modulating gamma amplitudes of 20 temporal-parietal electrodes (C3-6, T7-8, CP3-6, TP7-10, and P3-8) were calculated one by one to investigate the PAC temporal dynamics while participants performed a lengthy mental task (Results of the Wilcoxon test with $|Z| > 1.97$ and $p < 0.05$ were plotted in Fig. 3a). At all possible electrode couplings ($C_{13}^1 * C_{20}^1 = 260$), PAC in the heavier-load condition occurred more often than in the easier-load condition (All $F_{1/53} > 4.50$ and $p < 0.0386$, Fig. 3b). This result meant that more PACs were observed with increasing task difficulty.

In the first 40 min (time 1 to time 4), the maximal gamma amplitudes were locked at the trough of the theta cycle (ANOVA: all $F_{14/742} > 24.08$ and $p < 0.0001$). However, when the working time was extended to 50 min, 20 of 54 participants showed a changed PAC pattern in which gamma-frequency activity burst occurred slightly near the trough of the theta period (ANOVA: $F_{14/266} = 7.25$ and $p = 0.0010$), and other participants retained the original PAC pattern (ANOVA: $F_{14/462} = 18.42$ and $p < 0.0001$). This kind of PAC continued until 60 min, when gamma amplitudes of the same participants were locked near the trough (ANOVA: $F_{14/266} = 5.70$ and $p = 0.0046$) or at the trough (ANOVA: $F_{14/462} = 83.03$ and $p < 0.0001$) of the theta cycle. Then, the PAC of previous participants with the trough-locked amplitude changed dynamically when the WM task continued until 70 min, and all PAC changed to the near-trough locked pattern (ANOVA: $F_{14/742} = 17.46$ and $p < 0.0001$). Interestingly, 20 participants with earlier dynamics had low coupling if the working length was >70 min, and the PAC pattern was not locked at the trough or near the trough (ANOVA: all $F_{14/266} = 1.05$ and $p = 0.3991$). The quantitative preponderance of trough or near-trough coupling gradually disappeared and was replaced by atypical coupling toward the end of the 90 min lengthy mental task (ANOVA: $F_{14/742} = 0.94$ and $p = 0.5026$).

On average (Supplementary Fig. 1c), the accuracy rate decreased as the lengthy mental task extended from time 1 ($97.62\% \pm 1.86\%$) to time 9 ($96.39\% \pm 2.83\%$), and participants generally performed better in the easier-load condition

($97.59\% \pm 1.86\%$) than in the heavier-load condition ($96.42\% \pm 2.84\%$), as indicated by significant main effects of time (time 1 vs. time 9, $F_{1/53} = 16.90$ and $p = 0.0001$) and condition (easier load vs. heavier load, $F_{1/53} = 13.78$ and $p = 0.0005$) in the two-way ANOVA. Furthermore, a significant interaction was found for time × condition ($F_{1/53} = 14.83$ and $p = 0.0003$). One-way ANOVA showed no difference ($F_{1/53} = 0.076$ and $p = 0.7839$) in performance for time 1 between the easier-load condition ($97.56\% \pm 1.88\%$) and the heavier-load condition ($97.67\% \pm 1.86\%$), whereas for time 9 ($F_{1/53} = 21.26$ and $p < .0001$), performance declined by 2.44% in the heavier-load condition ($95.17\% \pm 3.10\%$) compared to that in the easier-load condition ($97.61\% \pm 1.86\%$). Furthermore, participants could be divided into two groups when thinking about the time during which PAC dynamically changed. Therefore, we tried to determine the differences between the two groups of participants. One interesting phenomenon was revealed in that these participants with faster adaptability (there was a practice stage before the task started, see methods section for details) to the length of the WM task showed earlier changes in PAC (one-way ANOVA: $F_{1/53} = 8.504$ and $p = 0.0052$, as shown in Fig. 3c). Although PAC temporal dynamics were asynchronous between the two groups, no significant accuracy difference in the lengthy mental task was shown at any time phase (one-way ANOVA: all $F_{1/53} < 3.366$ and $p > 0.0723$, as shown in Fig. 3d).

**A lengthy mental task did not alter the TWA increase in the FCC surface for the memorization condition.** TWA, as an EEG signature indicative of slow-wave activity, has been shown to correlate closely with a net synaptic strength marker in humans[53–55]. The full EEG spectrum demonstrated a more significant increase in theta power during the two-back tasks than during the resting conditions (Fig. 4a). EEG power in the 4–8 Hz (theta) frequency range of the I-2B task was higher than that of the corresponding resting condition ($T_{53} = -5.00$, $p < 0.0001$), and this kind of oscillation mechanism was also observed in the II-2B task ($T_{53} = -6.40$, $p < 0.0001$). Namely, the TWA in the FCC surface of the resting awake condition was significantly lower than that of the memorizing interval condition, indicating that increased cortical excitability at theta frequency would act as a visual indication for memory maintenance. Interestingly, the spectral power changes in TWA were independent of the lengthy mental task (Resting: $T_{53} = -0.10$, $p = 0.9187$; Memorizing: $T_{53} = -0.25$, $p = 0.8064$).

**Sleep deprivation would excessively increase TWA in the FCC surface and would not decouple theta-gamma PAC.** In the second kind of ground-based experiment before spaceflight, in which 38 participants experienced 2 days and one night of sleep deprivation. The results showed that whether participants experienced sleep deprivation, TWA observed during memorization was always significantly higher than the resting TWA, as shown on the full EEG spectrum (Fig. 4b). The EEG spectrum without sleep deprivation demonstrated a highly significant increase in theta power in the memorization condition compared to the theta power in the resting condition ($T_{37} = -5.86$, $p < 0.0001$), indicating memory maintenance, similar to the changing trend before participants performed the lengthy mental task. This significant power increase in the theta range continued to the end of the 36-h period of sleep deprivation ($T_{37} = -7.42$, $p < 0.0001$), while the mean accuracy rate of the two-back task was decreased significantly ($F_{1/37} = 256.76$, $p < 0.0001$, Supplementary Fig. 2a).

This result supported the conclusion that a relative increase in theta power from resting to memorizing conditions could act as a

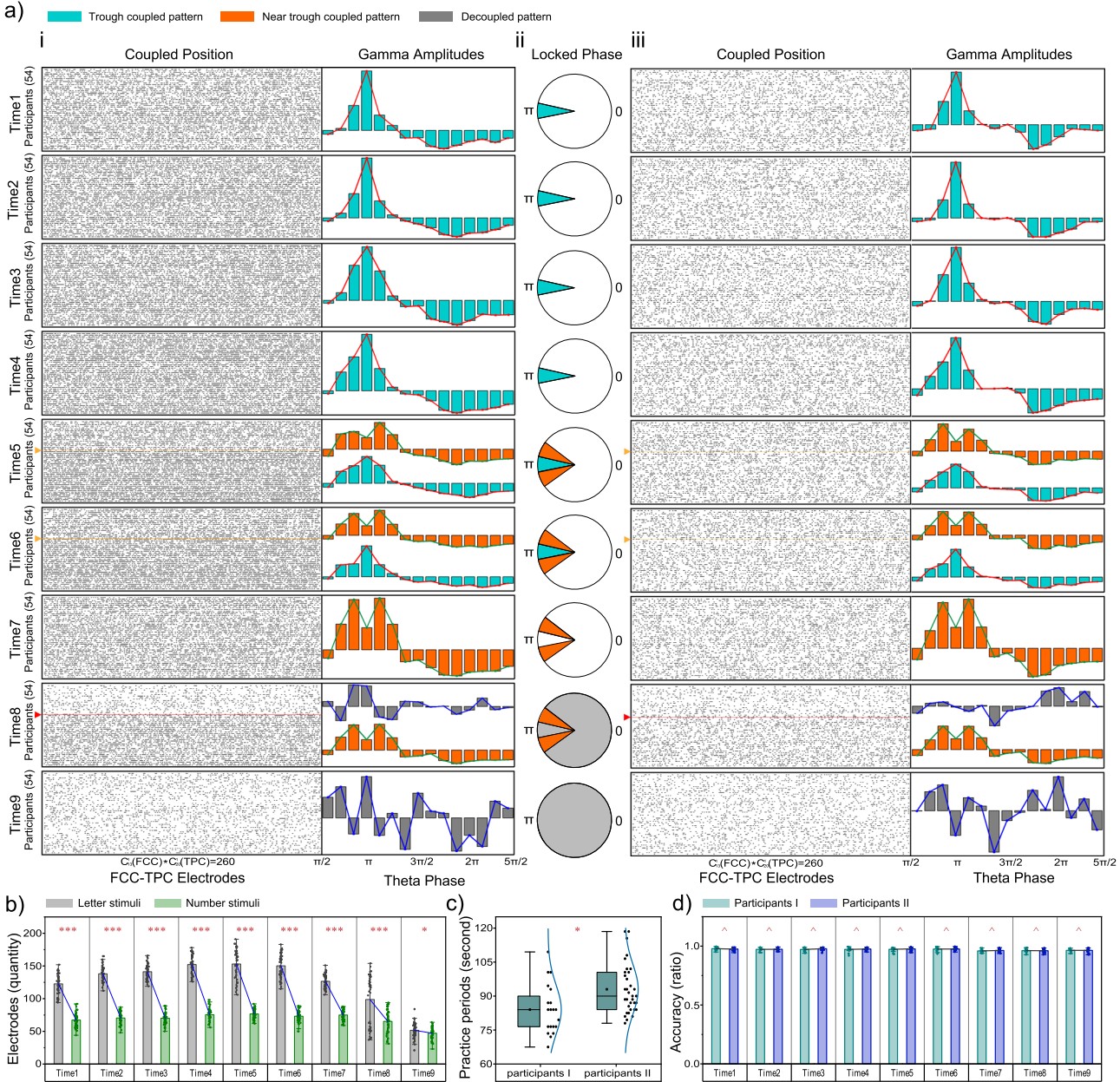

**Fig. 3 Dynamics of the theta-gamma phase-amplitude coupling.** $n = 54$, **a** Black dots indicated coupled positions, that theta phases of 13 electrodes in the frontal-central cortex (FCC) surface modulating gamma amplitudes of 20 electrodes in the temporal-parietal cortex (TPC) surface, for each participant, a total of 260 (13*20) electrode positions where theta-gamma phase-amplitude coupling (PAC) may occur. Cyan bars indicated that averaged gamma amplitudes in TPC surface were locked at the trough of the theta cycle in FCC surface. Orange bars indicated that averaged gamma amplitudes burst near the trough of the theta period. Grey bars showed that the coupled phase of the trough or near the trough disappeared. Polar coordinates in the ii panel of the figure illustrated positions of the coupled phase in a cosine function, in which $\pi$ was the minimum point. From time 1 to time 9, the number of coupled electrodes gradually decreased, and the position of coupled phase gradually moved away from the theta wave trough. Yellow and red lines divided participants into the I and II group, and they have different dynamics of the theta-gamma PAC. The (i) and (iii) panel showed PAC dynamics of the heavier-load and easier-load condition in the length mental task, respectively. **b** PAC of the heavier-load condition (gray columns, means ± standard deviations) occurred more often than the easier-load condition (green columns, means ± standard deviations), indicated by a one-way repeated ANOVA (***$p < 0.0001$). **c** The practice periods (means ± standard deviations) for the I and II group of participants was significant different, indicated by a one-way repeated ANOVA (*$p < 0.05$). **d** Behavior performance, the accuracy rate of the lengthy mental task, was not significant difference between I (cyan columns, means ± standard deviations) and II (blue columns, means ± standard deviations) group of participants, indicated by a one-way repeated ANOVA (^$p > 0.05$).

visual indication for memory maintenance[56]; however, this trend did not effectively indicate WM performance of participants who experienced sleep deprivation. Our data showed that the theta power density of sleep deprivation EEG was considerably higher than baseline EEG in the resting ($T_{37} = -5.53$, $p < 0.0001$) or

memorizing ($T_{37} = -5.75$, $p < 0.0001$) condition, demonstrating that sleep deprivation resulted in an excessive increase in net synaptic strength, especially in resting conditions. Fortunately, the theta-gamma PAC reflected between the FCC and TPC surface remained steady in 38 participants after sleep deprivation

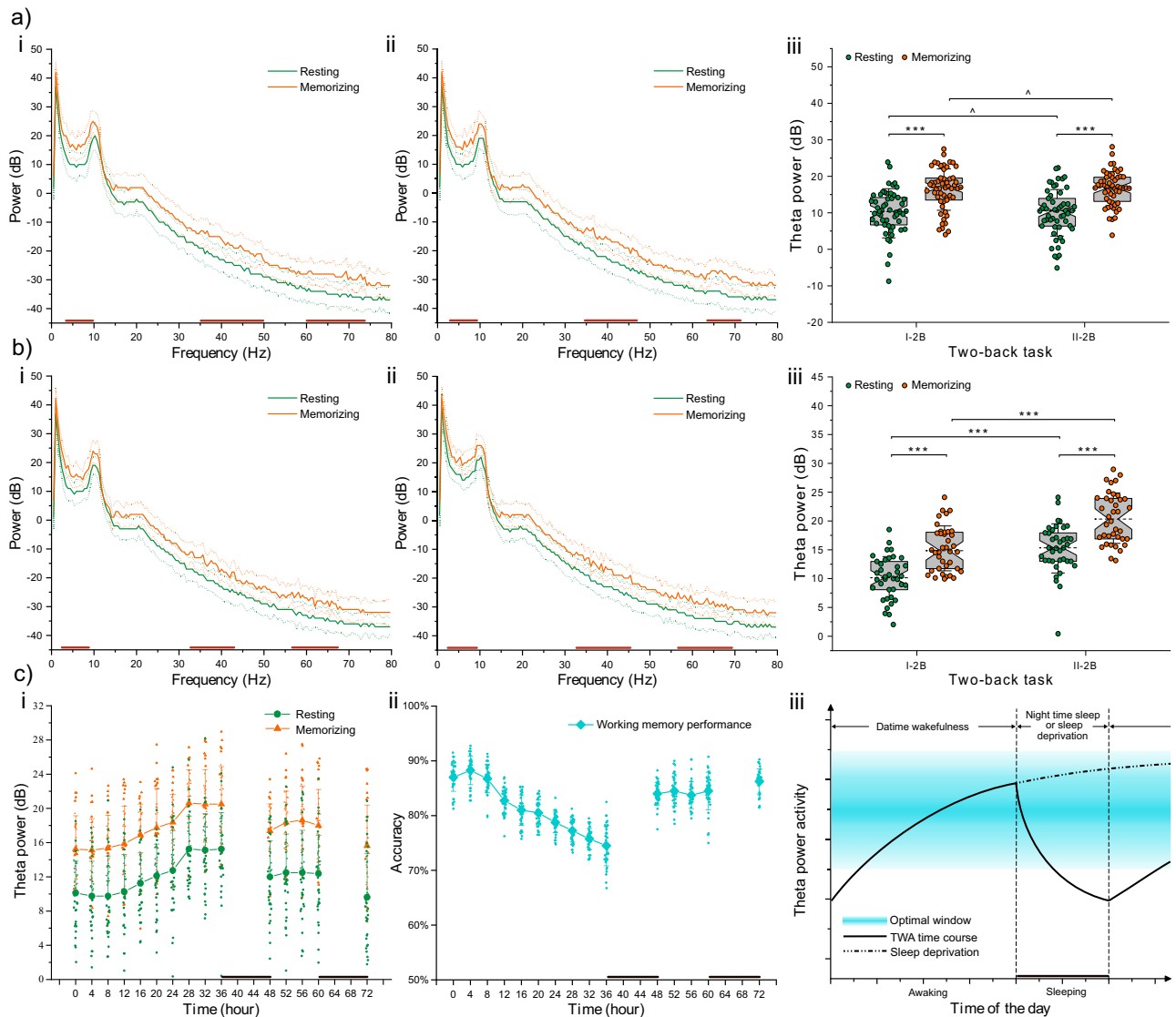

**Fig. 4 Analyses of the theta wave activity. a** $n = 54$, (i) and (ii) panel show the electroencephalographic (EEG) spectral power before and after the lengthy mental task, respectively. Solid lines represent means. Dashed lines represent 95% confidence intervals. Paired-sample $t$-tests of two-tailed were used to reveal statistical differences for all frequency ranges by the 1 Hz step. Theta power density in the 4–8 Hz frequency range of memorizing condition was significantly higher than that of the resting condition. For the frontal-central cortex (FCC) surface, (iii) panel shows the averaged theta wave activity (TWA) in the resting (green dots) and memorizing (orange dots) conditions. After the lengthy mental task, TWA in FCC surface was not significantly different (^$p > 0.05$). The whisker threads in both ends of the boxplot represent standard deviations, and dashed lines in the middle of the boxplot represent medians. **b** $n = 38$, i and ii panel show the EEG spectral power before and after sleep deprivation of 36 h, respectively. Solid lines represent means, and dashed lines represent 95% confidence intervals. Theta power density in the 4–8 Hz frequency range of memorizing condition was significantly higher than that of the resting condition. (iii) Panel shows the averaged resting (green dots) and memorizing (orange dots) TWA in FCC surface. After the sleep deprivation, TWA in FCC surface was significantly higher (***$p < 0.0001$). The whisker threads in both ends of the boxplot represent standard deviations, and dashed lines in the middle of the boxplot represent medians. **c** $n = 38$, (i) panel shows the TWA change curve with the sleep deprivation and recovery time course. Green and orange dots represent resting and memorizing conditions, respectively. Data is shown by means + standard deviations. (ii) Panel shows the behavior performance curve with the sleep deprivation and recovery time course. Data is shown by means + standard deviations in cyan rhombus. (iii) Panel depicts the proposed interplay between the time course of TWA (sleep-dependent homeostatic: solid and dotted lines) and the behavior performance of working memory (cyan window). Wakefulness is associated with an upscaling, while sleep is associated with a downscaling of TWA. Sleep deprivation eventually leads to TWA saturation and a behavioral obstacle to working memory.

(Supplementary Fig. 2b, c), providing the evidence for dissociating abnormal oscillation mechanisms caused by the lengthy mental task or sleep deprivation.

**Sleep deprivation gradually impeded the sleep-dependent recovery mechanism of TWA in the FCC surface.** Over time (Fig. 4c), the change curve for theta power density began to shift upward after ~16 h of continuous wakefulness of participants and

increased significantly (Resting: $T_{37} = -2.45$, $p = 0.0192$; Memorizing: $T_{37} = -2.37$, $p = 0.0229$) from 20 h compared to that at a baseline of 0 h. This upward trend lasted until 36 h, which confirmed that TWA was continuously increased at later stages of sleep deprivation, despite nearly saturated growth in the last hours of the sleep-deprived period. Interestingly, beginning soon after the first sleep recovery of 12 h, theta power was reduced compared with the highest density that was observed during the

period of sleep deprivation (Resting: $T_{37} = 2.97$, $p = 0.0052$; Memorizing: $T_{37} = 4.03$, $p = 0.0003$). Participants showed a restored TWA status after the second sleep period, which aligned to the baseline level (Resting: $T_{37} = 0.54$, $p = 0.5930$; Memorizing: $T_{37} = -0.42$, $p = 0.6785$), and demonstrated lower theta power than that observed after the first sleep recovery (Resting: $T_{37} = 2.38$, $p = 0.0225$; Memorizing: $T_{37} = 2.18$, $p = 0.0361$). Additionally, the mean accuracy rates of the two-back task were overall negatively correlated with theta power density. Thus, the recovery mechanism to theta power, damaged by sleep deprivation, depended on sufficient sleep, which might be essential for the synaptic homeostasis of memories.

**Theta-gamma PAC and TWA remained steady in the extreme nutritional supply or psychological situation conditions.** In this study, 25 healthy volunteers were recruited to fast completely for up to 7 days, and thirteen volunteers persisted until the end of this experiment. Nevertheless, theta-gamma PAC, gamma-amplitude modulation by theta phase activity, was still apparent between the FCC and TPC surface after participants fasted and when they performed the two-back task (Supplementary Fig. 3a); i.e., a burst of TPC gamma activity locked to the negative trough of FCC theta activity compared to shifted data (Most $|Z| > 1.97$ and $p < 0.05$, FDR-corrected, Supplementary Fig. 3b). In addition, fasting for 7 days did not affect the TWA in the FCC surface (Supplementary Fig. 3c); i.e., theta power increased significantly from the resting condition to the memorizing condition ($T_{12} = -7.85$, $p < 0.0001$). Moreover, TWA did not show a significant difference compared with baseline (Resting: $T_{12} = -0.98$, $p = 0.3463$; Memorizing: $T_{12} = -1.25$, $p = 0.2362$). Additionally, some representative moods occurred in our ground-based experiments with the lengthy mental task, sleep deprivation, and complete fasting, including 22 instances of excitements, 15 instances of anger, 14 instances of fear, 17 instances of sadness, 24 instances of anxiousness, 10 instances of hate, and 11 instances of depression. These data were utilized to demonstrate whether psychological situations affect theta-gamma PAC or theta power in the FCC surface and were excluded from the above analyses. This result showed that the decoupling of theta-gamma PAC and the change in frontal-central TWA were independent of psychological situations (Supplementary Figs. 4 and 5).

**The decoupled mechanism of theta-gamma PAC during WM tasks occurred in astronauts living in the China Space Station.** The neuronal oscillations of the cerebral cortex might differ between tasks with physical and mental loads; hence, data collected from astronauts after daily work with and without heavy mental loads were exhibited and analyzed separately (Fig. 5a, b). We found that 8/6 of 12 ground/space data points showed a robust theta-gamma PAC during the middle stages ($F_{7/98} = 23.43$ and $p < 0.0001/F_{5/70} = 8.76$ and $p = 0.0019$, Fig. 5a, b). Other data points ($N = 4/6$) were not statistically significant, reflecting that these theta-gamma modulations were decoupled (for all these data, the maximal gamma amplitudes were not locked near the trough of the theta cycle, $F_{3/42} = 1.62$ and $p = 0.2519/F_{5/70} = 0.77$ and $p = 0.5506$). However, during the end stages of the simulator/spaceflight period, the number of coupled data was decreased to 6/5 ($F_{5/70} = 12.21$ and $p = 0.0004/F_{4/56} = 9.76$ and $p = 0.0047$), and more decoupled PACs ($N = 6/7$) were detected ($F_{5/70} = 0.88$ and $p = 0.4842/F_{6/84} = 1.55$ and $p = 0.2162$). Notably, most decoupled PACs occurred after heavy mental work rather than after physical work (middle stages of simulator/spaceflight period: 3 vs 1/4 vs 2; end stages: 4 vs 2/5 vs 2).

We calculated the ratio of coupled electrodes to determine whether there was a difference in the degree of theta-gamma PAC

coupling when astronauts performed memory tests on the ground vs. space. This result indicated that fewer PACs were detected on the FCC and TPC surface of astronauts in the space station than on the ground (middle stages: $F_{(7, 5)} = 7.31$ and $p = 0.0192$; end stages: $F_{(5, 4)} = 24.38$ and $p = 0.0008$, Fig. 5c). Moreover, the TWA in the FCC surface of all astronauts who exhibited a decoupled mechanism of theta-gamma PAC did not differ significantly compared with that of other astronauts (resting: $T = 0.0007$, $p = 0.9995$; memorizing: $T = 0.1703$, $p = 0.8655$, Fig. 5d); in addition, its increase pattern remained apparent during WM performance (Coupled group: $T_{24} = -5.5466$, $p < 0.0001$; decoupled group: $T_{22} = -6.8896$, $p < 0.0001$). On the ground, no significant changes in TWA in the FCC were found between the two groups divided by whether PAC was decoupled (Resting: $T = -0.1308$, $p = 0.8971$; memorizing: $T = -0.0357$, $p = 0.9718$, Fig. 5e), and the TWA of the memorizing condition still increased significantly compared to the resting condition (Coupled group: $T_{13} = -4.609$, $p = 0.0005$; Decoupled group: $T_9 = -5.7707$, $p = 0.0003$). On the space station, changes in TWA in FCC surface were also not significant between them (resting: $T = 0.2663$, $p = 0.7925$; memorizing: $T = 0.5470$, $p = 0.5899$, Fig. 5f), along with increased TWA during the WM period (coupled group: $T_{10} = -3.2904$, $p = 0.0081$; decoupled group: $T_{12} = -4.5623$, $p = 0.0007$).

## Discussion

Previous studies[57] have indicated that the trough of slow-wave oscillations is associated with faster neuronal spiking than the more inhibitory peak, a principle also suggested to hold true for frontal theta waves as a component of cognitive control[58]. James et al.[59] reported that theta band activities over the mid-frontal cortex appeared to reflect a common computation used to achieve cognitive control; these theta oscillations might be used to communicate the need for cognitive control and subsequently implement such control across disparate brain regions as a biologically plausible candidate for neuronal communication. Consistent with the above studies, temporal-parietal gamma amplitude was nested around the frontal-central theta trough when participants were performing the WM task, a two-back letter task, in our experiments. Berger and colleagues[49] demonstrated that right temporoparietal high-frequency amplitudes were nested around the frontal-midline theta trough during a greater load WM task, and the nested phases changed depending on cognitive demand while disappearing when tasks were easier. Our results from the lengthy mental task, in which the heavier-load condition had more coupled EEG electrodes than the easier-load condition, provide additional evidence for the theory that the FCC exerts greater cognitive control in highly demanding WM tasks.

Moreover, the neuronal oscillation associated with frontal-central theta-phase modulation of temporal-parietal gamma activity is gradually decoupled over the lengthy mental task, even though the WM load remains the same. Given that the excitatory theta phase and increased neuronal activity indicated by gamma amplitudes are separated by up to nearly 100 ms, it is improbable that much neural communication between the frontal-central and temporal-parietal cortices takes place in this condition (Fig. 2e). Therefore, we propose that frontal-central cortical regions associated with cognitive control seem to passively desynchronize with the TPC by decoupling theta-gamma PAC in situations where excessive mental effort has been exerted such that the gating mechanism for administrating cognitive resources malfunctions because of long-term use. Researchers[60] have indicated that WM processes for a brain relying on theta/gamma band oscillations emerge from different functional networks involving

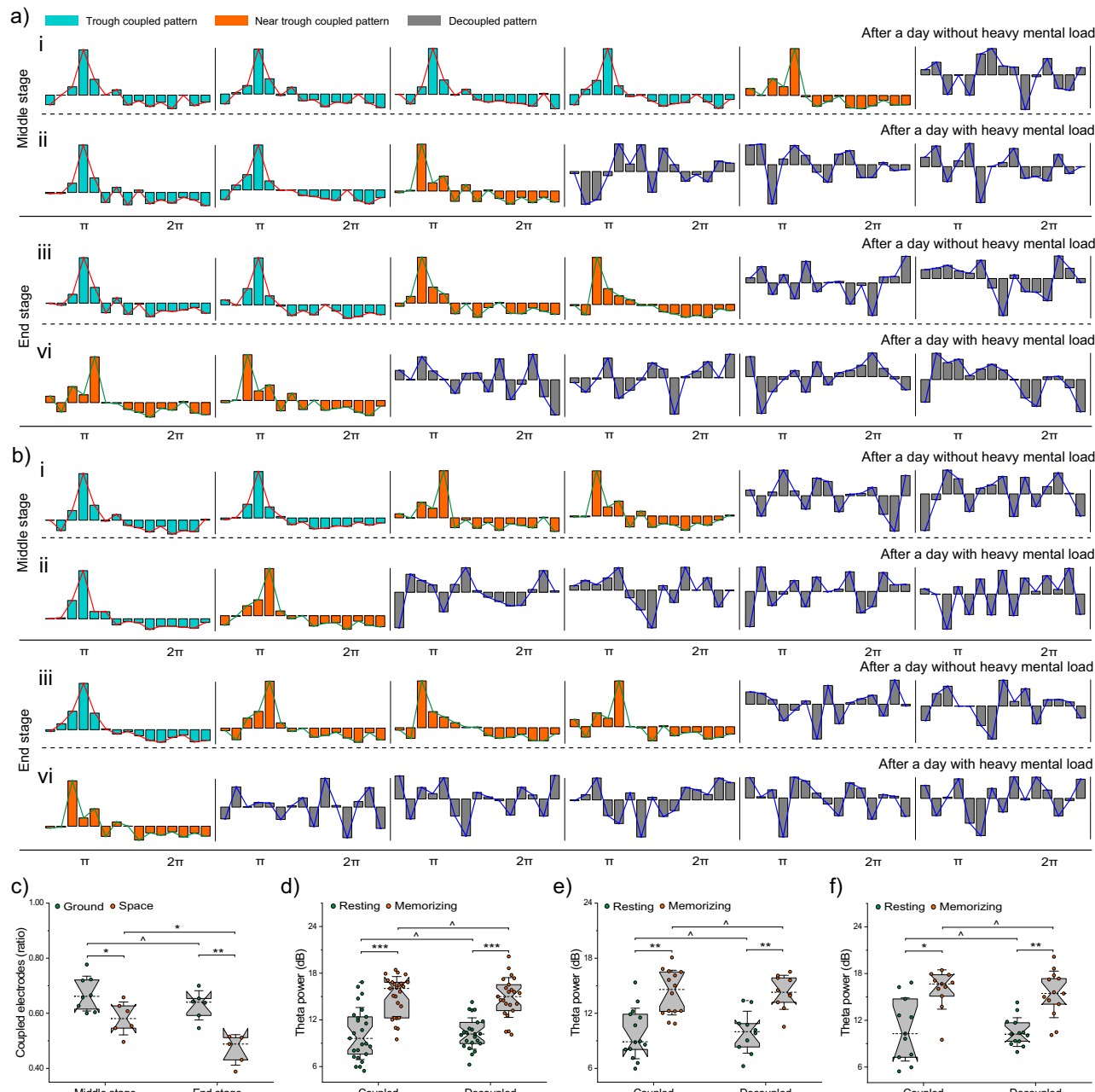

**Fig. 5 Test results of astronauts. a** and **b** Show the theta-gamma phase-amplitude coupling (PAC) or decoupled PAC of astronauts on the ground and the space station, respectively. Cyan bars indicated that averaged gamma amplitudes were locked at the trough of the theta cycle. Orange bars indicated that averaged gamma amplitudes burst near the trough of the theta period. Grey bars showed that the coupled phase of the trough or near the trough disappeared. (i) and (ii) Panel show tested results in the middle stage after a day without heavy mental load and with heavy mental load, respectively. (iii) and (iv) panel show tested results in the end stage after a day without heavy mental load and with heavy mental load, respectively. **c** Green dots and orange dots show the ratio of coupled electrodes in the middle ($n = 8$ vs. $n = 6$) and end ($n = 6$ vs. $n = 5$) stages, respectively. Data is shown by means ± standard deviations. Whether the ratio was significantly changed, indicated by the one-way ANOVA (^$p > 0.05$, *$p < 0.05$, **$p < 0.001$). **d** Green dots and orange dots show the averaged theta wave activity (TWA) of the resting and memorizing conditions in coupled group ($n = 25$), respectively. Data is shown by boxplot. The whisker threads in both ends of the boxplot represent standard deviations, and dashed lines in the middle of the boxplot represent medians. TWA of memorizing condition was significantly higher than that of the resting condition, indicated by the paired-sample $t$-tests of two-tailed (*$p < 0.05$, **$p < 0.001$, ***$p < 0.0001$). When the theta-gamma PAC was decoupled ($n = 23$), TWA was not significantly different (^$p > 0.05$). Data of astronauts (couped $n = 14$ vs. decoupled $n = 10$) before spaceflight are shown in figure (**e**) Data of astronauts (couped $n = 11$ vs. decoupled $n = 13$) in spaceflight are shown in figure (**f**).

multiple cortical areas. A study by Daume et al.[61] suggested an alternative mechanism by which frontal theta waves might control fast oscillatory activity in posterior brain regions during visual WM, and local coupling between theta phase and gamma amplitudes was observed in the inferior temporal cortex. In contrast, our data revealed abundant local PAC inside the bilateral temporal-parietal, occipital-parietal, or central-parietal cortices, while only FCC and TPC interregional PAC decoupled after the lengthy mental task. We speculate that the gating mechanism outlined above is relatively vulnerable in WM processes;

therefore, they need to be protected by countermeasures[62]. Of course, a limitation of these results described in this study is that all interpretations are based on the cortical areas where the surface EEG signal is detected, not the intracranial source of the signal.

Furthermore, Jakovcevski et al.[63] confirmed that WM function is impaired after neuronal ablation related to short-term synaptic plasticity in the prefrontal cortex of the mouse. Healthy participants in the current study show that frontal-central TWA, an index of net synaptic strength, instantaneously increases during memorization conditions compared to that in resting conditions. We propose that the frontal-central neuronal connection is temporarily strengthened during memory maintenance by relying on short-term synaptic plasticity to communicate with other brain regions; this occurrence can be shown by the TWA in FCC surface. In line with this mechanism of neuronal oscillations, Pugin et al.[64] reported that in vitro or in vivo animal experiments and studies of humans indicated that local slow-wave activity changes reflect synaptic plasticity, and slow-wave activity is increased in the frontoparietal cortex because of intensive WM training. Hosseinian and colleagues[4] further demonstrated that boosted frontal theta activity in humans is accompanied by enhancement of WM performance. Importantly, our data show that the frontal-central increased TWA elicited by WM is independent of lengthy mental task performance, which is different from the gating mechanism described above.

Although a lengthy period of mental work does not affect frontal-central TWA, we demonstrate a notable increase in frontal-central TWA after sleep deprivation in the human cortex, either for memorizing or resting conditions, and a behavioral deficit in performing the two-back task. These results are supported by Kuhn et al.[55], who also demonstrated significantly enhanced waking TWA and a deficit in the encoding of a word-pair task after sleep deprivation and suggested that partial occlusion in long-term synaptic plasticity after sleep deprivation can translate to the behavioral level. Furthermore, we demonstrated that task-related TWA is always higher than resting TWA regardless of whether participants experience sleep deprivation; moreover, and both TWAs recover to baseline levels after two 12-h periods of sleep recovery. Miyamoto et al.[26] confirmed that long-term potentiation of synaptic plasticity in the rat barrel cortex is not induced by electrical stimulation during sleep deprivation, and Kuhn et al.[55] confirmed decreased inducibility of associative synaptic long-term potentiation due to saturation after sleep deprivation. Thus, our data further provide evidence for the synaptic homeostasis hypothesis[22] that homeostatic modifications of net synaptic strength across the sleep-wake cycle define the set-point for the induction of synaptic plasticity. Together, these results point to a limited daily capacity for the cortex to undergo synaptic strength unless sleep occurs, while WM performance of humans decreases when TWA is beyond the limit of this optimal window (Fig. 4c-iii).

More recently, Chen B. et al.[65] detected more brain rhythm pairs of positive or negative correlation in their designed cognitive task, including θ-γ. We found that θ-β, α-β, and α-γ were also significant coupled in our designed WM tasks, but their coupled cortical locations were different from θ-γ (Supplementary Fig. 6). Moreover, only θ-γ decoupled after the lengthy mental task. Lin A. et al.[66] have demonstrated the presence of stable interaction patterns among brain rhythms in healthy subjects during sleep. Although EEG signals of subjects during sleep were not recorded in our study, during WM tasks, brain rhythm pairs, including θ-γ, θ-β, α-β, and α-γ, were still robust in subjects who experienced 36 h sleep deprivation. Liu KKL. et al.[67] reported that different brain areas exhibit different network dynamics of brain wave interactions to achieve differentiation in function during different sleep stages. In our study, we cannot rule out the possibility that long-term tasks or sleep deprivation may cause dynamic changes in other brain rhythm pairs. The reason may be that our designed experiments' severity of mental load or sleep discomfort was insufficient to disturb these robust PAC.

In conclusion, two extreme conditions, the lengthy mental tasks and sleep deprivation, affect neuronal oscillations of the cerebral cortex in different ways, implying their mutual non-interference. It is pivotal to mission success for astronauts to optimize cognitive performance in response to unanticipated situations[4]. Therefore, the causes of human cognitive decline in space stations should be understood to formulate effective countermeasures. Although EEG is an effective technique to study the functional system of the human brain, very few other EEG measurements in space have been conducted, especially when astronauts perform cognitive tasks. Using EEG recordings, Petit G. et al.[17] investigated sleep pressure markers during visual tasks in the wakefulness of five astronauts throughout their long-term space mission and provided evidence for increased sleep pressure with decreased alertness in astronauts on the International Space Station. Further, we provide the evidence for dissociation of abnormal oscillation mechanisms caused by the lengthy mental task from sleep deprivation, with particular relevance for the performance of astronauts in flight, where both sleep deprivation and high physical/mental loads are prevalent. However, other hazards, except for lengthy-period tasks, sleep discomfort, malnutrition, and abnormal mood researched in this study, may affect human cognitive and motor function during spaceflights, such as space-relevant flows of charged particles[68] and microgravity[69]. The interaction between these hazards makes it difficult for us to accurately separate the specific factors for cognitive function changes of astronauts because sometimes physiological state changes are not unique in flight. Such as Ivanov PC. et al.[70] studied sleep-wake transitions in cardiac dynamics from hours-long ECG recording of astronauts during long-term spaceflight, where no change in autonomic regulation of cardiac dynamics was found comparing terrestrial and space-flight data. Thus, the degree of theta-gamma PAC coupling when astronauts performed memory tests in space was significantly lower than tests on the ground during the same stage of experiment periods, but it can only demonstrate that long-term spaceflight and other hazards in the space station could worsen PAC decoupling evolution. Spaceflight is not the only necessary condition for desynchronizing this brain rhythm wave oscillation detected on the scalp surface, which can still be induced by the length of mental tasks performed on the ground. Along with extended space travel, further investigation is needed to address whether there are effective countermeasures for extreme conditions in spaceflight.

## Methods

**Participants**. EEG signals were synchronously recorded from healthy volunteers when they were performing the two-back letter task. All the participants were right-handed and free of any mental or somatic disorder or medication use. Participants were not affected by neurological or psychiatric disorders and had normal or corrected-to-normal vision. The participants were recruited from the community or a university and required a high school degree or above for inclusion in study. Individuals were also compensated for participation. All the participants signed written informed consent before the experiment. The experiments were approved by the local Ethics Committee at the School of Biological Science and Medical Engineering, Beihang University. In addition, all the participants maintained a regular sleep-wake schedule before the study, documented by sleep wristwatches (Huawei Watch D, CHN). The Profile of Mood State was used to assess the participants' current mood states just before and after the experiment. Notably, to analyze the neuronal oscillations in different psychological situations, if the subjects were in an abnormal mood before the experiment, we still asked them to perform a two-back task immediately, accompanied by EEG acquisitions. These participants with abnormal mood or poor sleep were invited to participate in subsequent experiments on another day.

**Two-back task**. Before and after all experiments described below, the participants performed a two-back task based on the random presentation of letters (Fig. 1a). This task started with the presentation of an introduction prompt, followed by a fixation cross that was shown for 5 s. Then, letters were presented to the subject sequentially, with a letter panel appearing every 1.5 s, i.e., the letter panel appeared for 0.5 s, and a blank panel was presented for 1 s. This design was consistent for every single trial. In addition, participants were asked to press the left mouse button using the right index finger when the current letter matched the letter presented two intervals (3 s) prior. Otherwise, participants were to press the right mouse button using the right middle finger. Each two-back task with a total time of 10 min comprised 200 target stimuli and 200 nontarget stimuli, and all stimuli were presented randomly. This task was controlled by the Presentation software platform (E-Prime v3.0, Psychology Software Tools, Inc., USA). All task material was presented on a 2k LCD monitor (Dell, Inc., USA) of 24 inches, with a screen resolution of 2560*1440. Participants sat in ergonomic chairs during the experiment and adjusted the height so that they could face the screen properly. Instructions were given to each participant before the formal task, and each participant was allowed to practice in tutorial mode until he or she became familiar with the task.

**Experiment 1—lengthy mental task**. Participants: We measured the scalp EEG signals of 60 healthy volunteers in the lengthy mental task. Four subjects were excluded from this analysis due to extreme scales assessed by the Profile of Mood State, and two subjects were also excluded from all analyses because they had too many blinks and horizontal eye movements. Ultimately, 54 participants were retained for statistical analysis in the lengthy mental task experiment (33 males, 21 females, age: 29.6 ± 7.1 years, age range: 21–49 years).

Experimental design: Subjects were not allowed to engage in excessive mental or physical work during the 48 h prior to this experiment. This task (Fig. 1a) started with the presentation of an introduction prompt, followed by a fixation cross that was shown for 5 s. Then, a letter or numeral was presented to the subject sequentially, with a letter/numeral panel appearing every 1.5 s, i.e., the letter/numeral panel appeared for 0.5 s, and a blank panel was presented for 1 s. This design was consistent for every single trial. Participants were asked to press the left mouse button using the right index finger when the current letter matched the letter presented two intervals (3 s) prior. Otherwise, participants pressed the right mouse button using the right middle finger. In addition, if the parity of the current numeral matched that of the numeral presented two intervals (3 s) prior, participants were asked to press the space bar on the keyboard. Otherwise, no reaction was needed. A lengthy mental task included nine blocks and lasted 90 min. Each 10 min block comprised 200 target stimuli and 200 nontarget stimuli, and all stimuli were presented randomly. A 2-min rest period between the two blocks was provided to participants, and they could move their body and sight range during this time but were not allowed to close their eyes for a long period. Instructions were given to each participant before they performed formal task, and participants were allowed to practice in tutorial mode until they became familiar with the task. The presentation platform and seat equipment of this task were in line with those of the two-back task described above.

**Experiment 2—sleep deprivation**. Participants: We measured the scalp EEG signals of 50 healthy volunteers in the sleep deprivation experiment (Fig. 1b). Ten subjects were excluded from this analysis due to extreme scales assessed by the Profile of Mood State, and two subjects were also excluded from all analyses because they had too many blinks and horizontal eye movements. Ultimately, 38 participants were retained for statistical analysis in the sleep deprivation experiment (24 males, 14 females, age: 28.0 ± 3.7 years, age range: 24–36 years).

Experimental design: Subjects were not allowed to travel across the Earth's meridian or work shifts for 60 days before the sleep deprivation experiment. Within 1 month before the experiment, subjects needed to go to bed between 22:00 and 0:00 while getting up between 6:00–8:00, and their sleep duration had to be at least 7 h. Subjects stopped taking caffeine, alcohol, tobacco, and drugs the week before the experiment began. In addition, siestas and vigorous physical activity were prohibited 24 h before formal participation. Only those who met these requirements could participate in the sleep deprivation experiment. On the first day of the experiment, subjects were required to arrive at the laboratory before 7:30 am. As the baseline data, the first test was conducted at 8:00 am. EEG signals were collected every 4 h for the next 36 h of sleep deprivation duration, including during the 5-min resting period and the two-back tasking state. The subjects then entered an 12-h sleep recovery period, and they still underwent EEG acquisitions every 4 h for 12 h after waking up. This session was followed by a second 12-h sleep recovery period, with a final EEG acquisition that was performed immediately after participants woke up. In addition, subjects were required to fill out a questionnaire about the Profile of Mood State before each EEG acquisition. During the rest time, subjects were kept awake under supervision, and they were allowed to engage in activities that were not stressful and exciting, such as reading, watching movies, surfing the internet, and chatting with others. The laboratory provided subjects with regular diets and fruit.

**Experiment 3—complete fasting**. Participants: We measured the scalp EEG signals of 25 healthy volunteers in the complete fasting experiment (Fig. 1c). Eight subjects were excluded from this analysis due to voluntary demission. In addition, four subjects with abnormal psychological situations were excluded from this analysis at the end of the complete fasting period. Ultimately, 13 participants were included in the statistical analysis of the complete fasting experiment (7 males, 6 females, age: 34.1 ± 7.7 years, age range: 27–51 years).

Experimental design: Subjects were required to have no severe and chronic medical histories, including diabetes mellitus, cancer, cardiovascular disease, metabolic diseases, and tobacco or alcohol dependence. Included participants did not have an eating disorder and ate three regular meals a day for 180 days before the complete fasting experiment. One month before the start of fasting, the subjects were not allowed to drink alcohol or eat other food that irritates the stomach, and overeating was forbidden. The subjects entered the experimental site 3 days in advance to go to the hospital for a comprehensive physical examination. This experiment consisted of 4 periods: a baseline period of 2 days, a fasting period of 7 days, a calorie restriction period (to prevent the impact of sudden overeating) of 3 days, and a recovery period of 4 days. Subjects practiced and performed the two-back task during the baseline period, and the first EEG data were collected. The subjects fasted for 7 days during the second period, meaning that oral intake consisted only of water ad libitum. EEG data were collected every 3 days in the fasting duration, including in a 5-min resting period and the two-back tasking state. The amount of food given to the subjects gradually increased to protect their digestive function during the third period, and they could return to regular eating habits in the recovery period. Data were collected once for each of the subjects during these two periods. In addition, subjects completed the Profile of Mood State questionnaire twice a day, and they performed the two-back task once a characteristic emotion appeared, accompanied by acquisitions of EEG. Subjects were free to move during the experiment except for other physiological examinations but could not engage in physical and mental work or stay up late. Subjects were monitored by medical staff throughout the experiment to ensure their absolute safety, and only after a physical examination during the recovery period could they leave the experimental site.

**Experiment 4—astronaut test**. Participants: We measured the scalp EEG of 9 astronauts (7 males, 2 female, age: 47.7 ± 6.3 years, age range: 41–57 years) who participated in recent Shenzhou missions (Fig. 1d). These astronauts entered the Chinese space station in three groups of three and were in orbit for at least 10 weeks. Spaceflight data of a female astronaut with long hair were excluded from this analysis due to high noise signals, but her data with low noise signals before spaceflight were retained.

Experimental design: In the Shenzhou series of missions, the astronauts' daily work content was planned, and their workload was heavy. The current research intended to analyze states of tiredness among astronauts, so all tests were scheduled in the afternoon before the astronauts finished work. Astronauts work in a simulated cabin for at least 3 months before the rocket launch, and their missions with physical or mental load were rehearsed in equivalent activity intensity to that in space station. In the middle (5th week) and end stages (10 week) of the simulated duration, astronauts performed a two-back task, accompanied by EEG acquisitions. Compared with ground tests, the test implemented on the China Space Station was also arranged in the middle and late stages of the Shenzhou missions. The first crew, all men, were tested twice during spaceflight (6 data were collected, 3 in the middle stage vs 3 in the end stage). However, the second crew included a woman whose signal was noisy due to her long hair and the limited experimental conditions in space, so the two male astronauts completed one additional test after completing the two planned tests (6 data were collected, 3 in the middle stage vs 3 in the end stage). The third crew, two men and one woman, were tested four during spaceflight (12 data were collected, 6 in the middle stage vs 6 in the end stage). Each test during spaceflight was completed with the cooperation of two astronauts who had received technical training. The experimental procedures were shown to the ground command center on a large screen. If problems were found, they were corrected immediately. All spaceflight data, including the task performance, 5 min resting EEG, and 10 min tasking EEG, were compressed, packaged and sent back to the ground command center for analysis.

**EEG acquisition and preprocessing**. EEG data were acquired with a 64-channel amplifier system (BrainAmp, Brain Products, GER), and those scalp electrodes were mounted according to the international 10-10 system against a nose reference. For the convenience of research, 64 electrodes were divided into 5 clusters, including the FCC (AFz, AF3-4, Fz, F1-4, FCz, and FC1-4), TPC (C3-C6, T7-8, CP3-6, TP7-10, and P3-8), central-parietal cortex (CPC: Cz, C1-2, CPz, CP1-2, Pz, and P1-2), occipital-parietal cortex (OPC: POz, PO3-4, PO7-8, Oz, and O1-2), and frontal-limbic cortex (Fp1-2, AF7-8, F5-8, FC5-6, and FT7-10). All electrodes were used in ground-based experiments, but only 13 FCC electrodes and 20 TPC electrodes were used in spaceflight experiments. The ground electrode was set at the forehead center, and impedances were kept below 20 kilohm. Signals were registered between 0.01 and 80 Hz with a sampling rate of 1000 Hz. EEG data were rereferenced offline to the average of all electrode channels. Then, the decomposition algorithm of independent component correlation was applied to remove

eye blinks and movements. Finally, residual artifacts from the eye and muscle were inspected and corrected manually. EEG analyses and codes were completed in MATLAB (v.2019b, MathWorks, USA) involving EEGLAB v.2019 with the current source density (CSD) toolbox.

Artifact Removal: Recently, Liu et al.[71] provided compelling evidence that blink-related processes are dynamically modulated not only to accommodate internal cognitive loading demands but also to address specific sensory processing requirements due to differences in the external environment. It is evident that not only at rest condition but also during a work task involving repeated reading, eye blinking occurs and is accompanied by an oscillatory activity[72]. In this study, astronauts performed WM tasks involving letter switching in microgravity conditions, where microsaccades are easily caused and capable of producing gamma activity[73]. Dimigen[74] demonstrated that correction could be strongly improved by training the ICA on optimally filtered data in which spike potentials were massively overweighted. Thus, we tracked eye movements (RED, SMI, GER) synchronously with EEG signals and employed an advanced ICA algorithm based on spike potentials overweighting, which was explored by Dimigen. Dimigen provided Matlab code in the published paper[74]. The statistical results show that the frequency of blink and saccade of astronauts in the space station is higher than that on the ground (Supplementary Fig. 7a), indicating that the quality of the signals measured in the Chinese space station was worse than that carried out on the ground. Furthermore, to assess the effectiveness of the ICA-based artifact removal procedure, quantitative analyses were performed to evaluate the changes in regional signal power before and after artifact rejection, referencing Liu et al.[72] Results showed that the power ratio was significantly reduced in the FCC and TPC regions following artifact removal, and there was no significant difference in the signals measured in the Chinese space station and on the ground (Supplementary Fig. 7b, c) after artifact removal. This result indicated that the properties consistent with ocular artifact had been successfully removed from the EEG data.

**Phase-amplitude coupling analyses.** Trials division: For PAC individual analyses, each EEG of one participant was segmented into epochs of 1500 ms for every task separately, comprising a 500 ms encoding target/nontarget and a retention interval of 1000 ms. These segments containing artifacts were removed from further analysis. The artifact-free trials of all correct and incorrect responses were used, and the PAC calculation described below was applied to the 1000 ms retention interval. The Laplacian CSD toolbox (implemented in MATLAB) was used to attenuate effects due to volume conduction on interregional phase synchronization and attenuate the spurious power effects and phase synchrony caused by microsaccadic eye movements[49,75]. The event-related potential was subtracted from each artifact-free single trial to prevent stimulus-evoked influence on oscillatory activity.

Phase-locked amplitude modulation: To calculate phase-locked amplitude modulation, complex Morlet wavelet filtering was used to decompose single trials. EEG data were filtered separately with low and high frequencies using a 1 Hz step for 1-30 Hz and a 10 Hz step for 30–80 Hz. Single-trial phases of 6 Hz theta waves at each electrode and single-trial amplitudes of 30, 40, 50, 60, and 80 Hz gamma waves at each electrode were estimated. Then, the Z-transformed gamma amplitudes of 30, 40, 50, 60, 70, and 80 Hz were sorted with respect to the instantaneous phase angle theta of 6 Hz and averaged into 15 phase bins, covering 24° of a theta cycle for each bin.

PAC analysis based on an electrode: For each electrode of an individual participant, to evaluate whether the theta phase significantly modulates gamma amplitude, the cosine function, which is at a frequency of 6 Hz and sampled at 55 Hz, has the same temporal resolution as the gamma averages, was cross-correlated with the theta phase-sorted gamma amplitude for each trial within every electrode, separately. Two full cosine periods were used for cross-correlations with phase-locked amplitudes in each retention interval of 1000 ms. Thus, for each trial, if the theta phase of one electrode did not modulate the gamma amplitude of another electrode, it would expect a flat cross-correlogram with coefficients close to zero. However, purposeful gamma-amplitude modulation by theta phase activity should result in a cross-correlogram with a clear peak different from zero. This cross-correlation analysis was applied to each trial separately, and the absolute maximum of the cross-correlogram was then used to measure the theta-phase modulation of the gamma amplitude. Next, the same procedures of the above cross-correlation approach were repeated for surrogate data in which the theta phases were randomly shifted in each trial. Theoretically, these shifted data should not have a higher absolute maximum of the cross-correlogram. Last, before the cross-correlograms of the absolute maxima of the original data were statistically compared to those of the shifted data, all absolute maxima were Fisher-Z transformed. In this statistical analysis, one-tailed FDR-corrected Wilcoxon tests were used to compare cross-correlation coefficients because of nonnormally distributed data even after Fisher-Z transformation. This method revealed which electrode theta phase modulated which electrode gamma amplitude for each participant.

PAC analysis based on participants: The phase-locked amplitude bins were averaged in all trials on one electrode for one participant, and the shifted data were processed in the same way. Then, the above cross-correlation approach was repeated in these averaged data, and the absolute maxima of the cross-correlogram

were calculated for each participant. After analysis of all participants in the experiment, Wilcoxon tests were used again to evaluate whether the theta phase of one electrode significantly modulates the gamma amplitude of another electrode. In this way, after analysis of all electrodes revealed which electrode clusters of theta-phase activity more often modulated which electrode clusters of gamma amplitude for a participant. Exploratory analysis revealed TPC electrode clusters showing gamma-amplitude modulation by the FCC theta phase, but this modulation was attenuated after a lengthy mental task. Therefore, in the next step, analyses were focused on these two clusters, and one-way repeated-measures ANOVA, using PHASE BIN (segment 1-15: each bin covering 24°) as a factor and the FCC theta phase-sorted gamma amplitude within the TPC electrode clusters as the dependent variable, was applied.

**TWA analyses.** The full EEG spectrum was calculated by Periodogram's method of spectral estimation. Then, the averaged theta power density in the 4–8 Hz frequency range was described as TWA[55]. The resting EEG was performed for a 5 min period of sustained wakefulness for each test, and the average spectral power in the theta-frequency band of artifact-free 1000 ms epochs was calculated. The average spectral power was calculated for task EEG signals in a retention interval of 1000 ms. Especially when analyzing the TWA in task conditions, the event-related potential was subtracted from each artifact-free trial to prevent stimulus-evoked influence on theta oscillations[75].

**Statistics and reproducibility.** One-way and two-way ANOVA with correction for multiple comparisons (Sidak's procedure) was used to determine the significance of the difference between the two groups. $p$-values < 0.05 were considered to be statistically significant. The sample size used in this study is more than three and is also given in the figure legend in detail.

**Reporting summary.** Further information on research design is available in the Nature Research Reporting Summary linked to this article.

## Data availability
The raw data used to generate Figs. 2–5 are shown in Supplementary Data 1–4, respectively. All original data are available from the corresponding authors upon reasonable request.

## Code availability
We used publicly available software for analysis in this study. Here, we list the URLs for the software where details about them, including the computer codes: EEGLAB v.2019 (https://www.sccn.ucsd.edu/eeglab/download.php), Matlab Wavelet Toolbox (https://www.mathworks.com/products/wavelet.html), Matlab Script for Artifact Removal (www.github.com/olafdimigen/opticat).

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

## Acknowledgements

This work is supported by funding from the China Manned Space Medical Experiment Project (HYZHXM03003). We thank all astronauts on two recent Shenzhou missions for their contributions to the data acquisitions, P. Sauseng for assistance in EEG processing. We would also like to thank H.Y., who shared the eye-tracking data collected in sync with the EEG.

## Author contributions

P.Z. conceptualization, methodology, investigation, software, formal analysis, visualization, writing-original draft, writing-review and editing, validation, data acquisition, and data curation; J.Y. conceptualization, resources, writing-review and editing; Z.L. conceptualization, methodology, software, resources, supervision, project administration; H.Y. conceptualization, investigation, data acquisition, supervision, funding acquisition; R.Z. investigation, data acquisition, validation, supervision, funding acquisition; Q.Z. conceptualization, methodology, writing-review and editing, coordination, supervision, validation, funding acquisition.

## Competing interests

The authors declare no competing interests.
