## [Peer Review File · Communications Biology]

Reviewers' comments:

Reviewer #1 (Remarks to the Author):

The authors present a detailed analysis of rhythmical oscillations of neural populations reflecting working memory performance under extreme conditions (lengthy-period tasks, sleep deprivation, total fasting, heavy physical and mental workload for astronauts on the ground in a simulated cabin and in space). Phase-amplitude coupling (PAC) analysis is performed to quantify synchronization in the dynamics of theta and gamma cortical rhythms across cortical locations representing different functional modalities to access cognitive performance and the effects of stress and mental fatigue on memory.

This is a very interesting study, based on unique data (including data from astronauts on the ground and during spaceflight) and diverse experimental protocols, with results confirmed on several different population groups. Notably, the authors demonstrate that while sleep deprivation does not affect the theta-gamma PAC, the synchronization of these two cortical rhythms across frontal central cortex (FCC) and temporal parietal cortex (TPC) areas is lost with lengthy mental tasks. This provides first evidence for dissociation of abnormal oscillation mechanisms caused by the lengthy mental task from sleep deprivation, special relevance for the performance of astronauts in flight, where both sleep deprivation as well as high physical/mental loads are prevalent.

Synchronization phenomena between theta and gamma cortical rhythms have long been reported in the literature in relation to memory, however, to my knowledge, this is the first study that shows in detail the evolution in time from synchronized to desynchronized behavior with accumulation of memory fatigue during lengthy mental tasks. The reported results are novel, the analytical work and statistical tests (ANOVA repeated measures, Wilcoxon test, etc.) are adequate and support the conclusions.

To better assess the effect of accumulated cognitive fatigue in the astronauts population it would be instructive to present the results based on the structure of the experimental protocol – one set of figures showing results of astronauts tests in the middle of the ground simulator period vs the end of the period, as well as for the middle of the flight period vs the end of the flight period. It would be essential to show whether the degree of PAC changes with accumulated physical and mental stress during long-term simulator and in-flight periods.

The grouping of astronauts subjects shown in Figure 5 d–e, as well as in Figure 5 g–h is arbitrary, where some astronauts exhibit theta-gamma PAC synchronization and others don't under the same conditions (i.e., working memory tests on the ground vs tests during spaceflight). This does not allow for consistent physical interpretation of the working memory test results. If the hypothesis and the aim of the study is to show that the heavy memory load during astronauts day affects their cognitive task performance as reflecting by the degree of theta-gamma PAC, then the authors should separate astronauts who performed the memory task after a day of high physical activity without mental load vs astronauts who performed the memory task after a day without (minimal) physical activity and heavy mental load.

The authors focus on the theta-gamma coupling which is a traditional approach in the context of memory and cognition. However, recent studies have demonstrated that synchronous coordination and coupling of all cortical rhythms is essential in facilitating various physiological states (wake/sleep) and behaviors (physical/cognitive), and that distinct network of dynamic coupling among all cortical rhythms (even within a single cortical location, EEG-channel) are an essential hallmark of states and conditions. See for example: Chen B. et al. Ensemble of coupling forms and networks among brain rhythms as function of states and cognition. *Communications Biology*, 2022; 5: 82; Lin A. et al. Dynamic network interactions among distinct brain rhythms as a hallmark of physiologic state and function. *Communications Biology*, 2020; 3: 197. Liu KKL. et al. Plasticity of brain wave network

interactions and evolution across physiologic states. *Frontiers in Neural Circuits*, 2015; 9:62. It would be instructive if the authors show results or comment on how other pairs of cortical rhythms within or across cortical locations are affected by long-term memory tasks.

Regarding microgravity effects on brain dynamics and autonomic regulation in astronauts during spaceflight, the authors may discuss their findings in the context of pioneering studies on sleep-wake transitions in cardiac dynamics from hours-long ECG recording of astronauts during long-term spaceflight, where no change in autonomic regulation of cardiac dynamics was found comparing terrestrial and spaceflight data – see for example Ivanov PC. et al. Sleep-wake differences in scaling behavior of the human heartbeat: analysis of terrestrial and long-term space flight data. *Europhysics Letters* 1999;48:594-600. It would be essential if the authors could quantify whether there is a difference in the degree of theta-gamma PAC coupling when astronauts performed memory tests on the ground vs space.

Reviewer #2 (Remarks to the Author):

This study represents important work done in part on the Chinese space station where brain electrical activity was recorded by multiple EEG during a cognitive task involving working memory (WM, the two-back letter task).

Given the small number of studies that have recorded dynamic EEG activity in weightlessness and during cognitive tasks, these previous publications deserve to be cited and discussed in relation to the present study.

The manuscript is well written but suffers from an accumulation of multiple concepts including in detail the various factors that may occur not only in relation to working in weightlessness but also in the conditions of Earth's gravity. All this description ultimately giving the impression that what happens in microgravity is only an extension of what would happen on land if the workload were increased. The choice to favor a PAC analysis centered on theta and gamma rhythms limits the neuroscientific scope of this study. In particular, the advantage of having used a system of 62 surface electrodes has been neglected and only some of these electrodes have been chosen according to predetermined theoretical criteria. The major problem which then arises is that throughout the manuscript the authors use scientific elements previously demonstrated as coming from a neuroanatomically and physiologically identified regions of the brain to interpret the signals measured independently by surface electrodes.

On the contrary, it would have been more advantageous to carry out an analysis of the sources of these activities and not be satisfied with considering that the activities measured by a surface electrode had to correspond to a generator located under this same electrode, as this could lead to abusive interpretations.

Nowhere (except that eye and blink are removed by ICA) is there any mention of the importance of eye movements and blinks in relation to the oscillations present on the surface of the scalp. It is however obvious that not only at rest condition but also during a work task involving repeated letter reading, eye blinking occurs and is accompanied by an oscillatory activity that can increase the GFP by two to four times for frequencies ranging from 0.5 Hz to 4 Hz and this for an interval ranging from - 250 ms to 750 ms on either side of the blink (Liu et al., 2017). In fact, even if we remove the artefactual component of the blinks, the internal dynamics of the brain producing these events essential to the information processing cannot be neglected. Similarly, microsaccades are capable of producing gamma activity and it is very difficult to identify them, especially in microgravity.

In summary, if the quality of the signals measured in the Chinese space station is comparable to that carried out on the ground, which an analysis of the SNR in the two situations should demonstrate, these data deserve to be more exploited, by researching with broader perspective and more powerful tools for analyzing the specificity of cerebral functioning in weightlessness.

Referee comments and responses checklist

We replied to all referee comments within the table below. Each reply is presented together with the relevant referee comment. Comments are numbered in one column and duplicated in the second column. Responses are listed in the third column, and corresponding changes to the manuscript/supplementary file are indicated by line number (such as 10-11/S:10-11) in the last column.

Number	Comment	Response	Location
Reviewer #1:			
1	The authors present a detailed analysis of rhythmical oscillations of neural populations reflecting working memory performance under extreme conditions (lengthy-period tasks, sleep deprivation, total fasting, heavy physical and mental workload for astronauts on the ground in a simulated cabin and in space). Phase-amplitude coupling (PAC) analysis is performed to quantify synchronization in the dynamics of theta and gamma cortical rhythms across cortical locations representing different functional modalities to access cognitive performance and the effects of stress and mental fatigue on memory.	We thank the reviewer for commenting on our research content.	/
2	This is a very interesting study, based on unique data (including data from astronauts on the ground and during spaceflight) and diverse experimental protocols, with results confirmed on several different population groups. Notably, the authors demonstrate that while sleep deprivation does not affect the theta-gamma PAC, the synchronization of these two cortical rhythms across frontal central cortex (FCC) and temporal parietal cortex (TPC) areas is lost with lengthy mental tasks. This provides first evidence for dissociation of abnormal oscillation mechanisms caused by the lengthy mental task from sleep deprivation, special relevance for the performance of astronauts in flight, where both sleep deprivation as well as high physical/mental loads are prevalent.	We thank the reviewer for approving our research.	/
3	Synchronization phenomena between theta and gamma cortical rhythms have long been reported in the literature in relation to memory, however, to my knowledge, this is the first study that shows in detail the evolution in time from synchronized to desynchronized behavior with accumulation of memory fatigue during	We thank the reviewer for approving this study's novelty, analytical work, and statistical tests.	/

Number	Comment	Response	Location
	lengthy mental tasks. The reported results are novel, the analytical work and statistical tests (ANOVA repeated measures, Wilcoxon test, etc.) are adequate and support the conclusions.		
4	To better access the effect of accumulated cognitive fatigue in the astronauts population it would be instructive to present the results based on the structure of the experimental protocol - one set of figures showing results of astronauts tests in the middle of the ground simulator period vs the end of the period, as well as for the middle of the flight period vs the end of the flight period. It would be essential to show whether the degree of PAC changes with accumulated physical and mental stress during long-term simulator and inflight periods.	We thank the reviewer for this comment that we should present the results based on the structure of the experimental protocol. Figure 5 a-b of the revised manuscript shows the results of astronauts' tests in the middle of the ground simulator period vs the end of the period, as well as for the middle of the flight period vs the end of the flight period. In addition, since the original manuscript was submitted to the present, we have completed a test of the other three-person flight crew on the Chinese space station; corresponding results were added to the revised manuscript. These analyses confirmed that the theta phase detected in the frontal-central cortex (FCC) surface still modulated gamma amplitude in the temporal-parietal cortex (TPC) surface when astronauts memorized after daily tasks, but sometimes this mechanism decoupled, regardless of whether people were on the ground or in orbit. Significantly, during the end stages of the simulator or spaceflight period, the number of coupled data was decreased, and more decoupled PACs were detected. Moreover, the ratio of coupled electrodes detected in astronauts with heavy mental loads was significantly lower in the end stage than in the middle stage (Fig. 5-c). These results show that the degree of PAC declined with accumulated physical and mental stress during long-term simulator and inflight periods.	522-564; 575-582; 945-948; 956; 960; 963-965;
5	The grouping of astronauts subjects shown in Figure 5 d-e, as well as in Figure 5 g-h is arbitrary, where some astronauts exhibit theta-gamma PAC synchronization and others don't under the same conditions (i.e., working memory tests on the ground vs tests during spaceflight). This does not allow for consistent physical interpretation of the working memory test results. If the hypothesis and the aim of the study is to show that the heavy memory load during astronauts day affects their cognitive task	We thank the reviewer for this comment that we should separate astronauts into the same conditions. The neuronal oscillations of the cerebral cortex might differ between tasks with physical and mental loads; hence, data collected from astronauts after daily work with and without heavy mental loads were exhibited and analyzed separately in the revised manuscript (Fig. 5a-b). After the daily work, some astronauts reflected that theta-gamma modulations were	522-564; 575-582;

Number	Comment	Response	Location
	performance as reflecting by the degree of theta-gamma PAC, then the authors should separate astronauts who performed the memory task after a day of high physical activity without mental load vs astronauts who performed the memory task after a day without (minimal) physical activity and heavy mental load.	decoupled (i.e., the maximal gamma amplitudes were not locked near the trough of the theta cycle). Notably, most decoupled PACs occurred after heavy mental work rather than after physical work (middle stages of simulator/spaceflight period: 3 vs 1 / 4 vs 2; end stages: 4 vs 2 / 5 vs 2). This result indicates that the degree of PAC decoupling with accumulated mental stress was more severe than with accumulated physical stress on the ground or in orbit.	
6	The authors focus on the theta-gamma coupling which is a traditional approach in the context of memory and cognition. However, recent studies have demonstrated that synchronous coordination and coupling of all cortical rhythms is essential in facilitating various physiological states (wake/sleep) and behaviors (physical/cognitive), and that distinct network of dynamic coupling among all cortical rhythms (even within a single cortical location, EEG-channel) are an essential hallmark of states and conditions. See for example: Chen B. et al. Ensemble of coupling forms and networks among brain rhythms as function of states and cognition. Communications Biology, 2022; 5: 82; Lin A. et al. Dynamic network interactions among distinct brain rhythms as a hallmark of physiologic state and function. Communications Biology, 2020; 3: 197. Liu KKL. et al. Plasticity of brain wave network interactions and evolution across physiologic states. Frontiers in Neural Circuits, 2015; 9:62. It would be instructive if the authors show results or comment on how other pairs of cortical rhythms within or across cortical locations are affected by long-term memory tasks.	We thank the reviewer for suggesting these more advanced research results about synchronous coordination and coupling of all cortical rhythms. Their results have broadened our research better to understand the relationship between brain rhythms and physiologic states. Such as, Chen B. et al. detected more brain rhythm pairs of positive or negative correlation in their designed cognitive task, including θ-γ. So that we further calculated all possible PAC pairs ($n = 10$) of the slow-wave phase and fast-wave amplitude between the δ, θ, α, β, and γ frequency range. We found that θ-β, α-β, and α-γ were also significant coupled in our designed WM tasks, but their coupled cortical locations were different from θ-γ (Fig. S6). Moreover, only θ-γ decoupled after the lengthy mental task. Lin A. et al. have demonstrated the presence of stable interaction patterns among brain rhythms in healthy subjects during sleep. Although EEG signals of subjects during sleep were not recorded in our study, during WM tasks, brain rhythm pairs, including θ-γ, θ-β, α-β, and α-γ, were still robust in subjects who experienced 36 h sleep deprivation. Liu KKL. et al. reported that different brain areas exhibit different network dynamics of brain wave interactions to achieve differentiation in function during different sleep stages. In our study, we cannot rule out the possibility that long-term tasks or sleep deprivation may cause dynamic changes in other brain rhythm pairs. The reason may be that our designed experiments' severity of mental	172-173; 778-792; S: 57-60;

Number	Comment	Response	Location
		load or sleep discomfort was insufficient to disturb these robust PAC. We have provided these results with supplementary materials and discussed them in the revised manuscript.	
7	Regarding microgravity effects on brain dynamics and autonomic regulation in astronauts during spaceflight, the authors may discuss their findings in the context of pioneering studies on sleep-wake transitions in cardiac dynamics from hours-long ECG recording of astronauts during long-term spaceflight, where no change in autonomic regulation of cardiac dynamics was found comparing terrestrial and spaceflight data - see for example Ivanov PC. et al. Sleep-wake differences in scaling behavior of the human heartbeat: analysis of terrestrial and long-term space flight data. Europhysics Letters 1999;48:594-600. It would be essential if the authors could quantify whether there is a difference in the degree of theta-gamma PAC coupling when astronauts performed memory tests on the ground vs space.	We thank the reviewer for this comment that we should discuss our findings in the context of pioneering studies. 1. It is pivotal to mission success for astronauts to optimize cognitive performance in response to unanticipated situations (Romanella S. et al., 2020). Therefore, the causes of human cognitive decline in space stations should be understood to formulate effective countermeasures. Although EEG is an effective technique to study the functional system of the human brain, very few other EEG measurements in space have been conducted, especially when astronauts perform cognitive tasks. Their main aim was to investigate the impact of microgravity on spontaneous brain oscillations and their modulation with a change of conditions from eyes open to eyes closed (Chiron et al., 2006), and during a visuo-attentional task preceding a visuo-motor docking task (Cebolla et al., 2016). More recently, EEG time series have been used to monitor the anomalous long-term effects in astronauts (Sommariva S. et al., 2022). Using EEG recordings, Petit G. et al. (2019) investigated sleep pressure markers during visual tasks in the wakefulness of five astronauts throughout their long-term space mission and provided first evidence for increased sleep pressure with decreased alertness in astronauts on the International Space Station. Further, we provide the first evidence for dissociation of abnormal oscillation mechanisms caused by the lengthy mental task from sleep deprivation, with particular relevance for the performance of astronauts in flight, where both sleep deprivation and high physical/mental loads are prevalent. However, other hazards, except for lengthy-period tasks, sleep discomfort, malnutrition, and abnormal mood researched in this study, may affect human cognitive and motor function	1. 797-828; 2. 17-19; 542-558; 575-582; 615-617;

Number	Comment	Response	Location
		during spaceflight. For example, cognitive deficits seem to result from a reduction of dendritic complexity and spine density due to exposure to space-relevant flows of charged particles (Parihar et al., 2015); microgravity seems to be associated with motor impairment as a consequence of the cortical reorganization of motor cortices (Demertzi et al., 2016). The interaction between these hazards makes it difficult for us to accurately separate the specific factors for cognitive function changes of astronauts because sometimes physiological state changes are not unique in flight. Such as Ivanov PC. et al. (1999) studied sleep-wake transitions in cardiac dynamics from hours-long ECG recording of astronauts during long-term spaceflight, where no change in autonomic regulation of cardiac dynamics was found comparing terrestrial and spaceflight data. Thus, the degree of theta-gamma PAC coupling when astronauts performed memory tests in space was significantly lower than tests on the ground during the same stage of experiment periods, but it can only demonstrate that long-term spaceflight and other hazards in the space station could worsen PAC decoupling evolution. Space flight is not the only necessary condition for desynchronizing this brain rhythm wave oscillation detected on the scalp surface, which can still be induced by the length of mental tasks performed on the ground. We have discussed them in the revised manuscript. 2. We calculated the ratio of coupled electrodes to determine whether there was a difference in the degree of theta-gamma PAC coupling when astronauts performed memory tests on the ground vs. space. This result indicated that fewer PACs were detected in the space station than on the ground (Fig. 5-c). We have provided these results in the revised manuscript.	
Reviewer #2:			
1	This study represents important work done in part on the Chinese space station where brain electrical activity was recorded by multiple EEG during a cognitive task	We thank the reviewer for commenting on our research content.	

Number	Comment	Response	Location
	involving working memory (WM, the two-back letter task).		
2	Given the small number of studies that have recorded dynamic EEG activity in weightlessness and during cognitive tasks, these previous publications deserve to be cited and discussed in relation to the present study.	We thank the reviewer for this comment that we should cite and discuss these previous publications. It is pivotal to mission success for astronauts to optimize cognitive performance in response to unanticipated situations (Romanella S. et al., 2020). Therefore, the causes of human cognitive decline in space stations should be understood to formulate effective countermeasures. Although EEG is an effective technique to study the functional system of the human brain, very few other EEG measurements in space have been conducted, especially when astronauts perform cognitive tasks. Their main aim was to investigate the impact of microgravity on spontaneous brain oscillations and their modulation with a change of conditions from eyes open to eyes closed (Chiron et al., 2006), and during a visuo-attentional task preceding a visuo-motor docking task (Cebolla et al., 2016). More recently, EEG time series have been used to monitor the anomalous long-term effects in astronauts (Sommariva S. et al., 2022). Using EEG recordings, Petit G. et al. (2019) investigated sleep pressure markers during visual tasks in the wakefulness of five astronauts throughout their long-term space mission and provided first evidence for increased sleep pressure with decreased alertness in astronauts on the International Space Station. Further, we provide the first evidence for dissociation of abnormal oscillation mechanisms caused by the lengthy mental task from sleep deprivation, with particular relevance for the performance of astronauts in flight, where both sleep deprivation and high physical/mental loads are prevalent. However, other hazards, except for lengthy-period tasks, sleep discomfort, malnutrition, and abnormal mood researched in this study, may affect human cognitive and motor function during spaceflight. For example, cognitive deficits seem to result from a reduction of dendritic complexity and spine density due to exposure to space-	797-828;

Number	Comment	Response	Location
		relevant flows of charged particles (Parihar et al., 2015); microgravity seems to be associated with motor impairment as a consequence of the cortical reorganization of motor cortices (Demertzi et al., 2016). The interaction between these hazards makes it difficult for us to accurately separate the specific factors for cognitive function changes of astronauts because sometimes physiological state changes are not unique in flight. Such as Ivanov PC. et al. (1999) studied sleep-wake transitions in cardiac dynamics from hours-long ECG recording of astronauts during long-term spaceflight, where no change in autonomic regulation of cardiac dynamics was found comparing terrestrial and spaceflight data. Thus, the degree of theta-gamma PAC coupling when astronauts performed memory tests in space was significantly lower than tests on the ground during the same stage of experiment periods, but it can only demonstrate that long-term spaceflight and other hazards in the space station could worsen PAC decoupling evolution. Space flight is not the only necessary condition for desynchronizing this brain rhythm wave oscillation detected on the scalp surface, which can still be induced by the length of mental tasks performed on the ground. We have cited and discussed these results in the revised manuscript.	
3	The manuscript is well written but suffers from an accumulation of multiple concepts including in detail the various factors that may occur not only in relation to working in weightlessness but also in the conditions of Earth's gravity. All this description ultimately giving the impression that what happens in microgravity is only an extension of what would happen on land if the workload were increased.	We thank the reviewer for commenting on our research content. Figure 5 a-c of the revised manuscript shows the results of astronauts' tests in the middle of the ground simulator period vs the end of the period, as well as for the middle of the flight period vs the end of the flight period. In addition, since the original manuscript was submitted to the present, we have completed a test of the other three-person flight crew on the Chinese space station; corresponding results were added to the revised manuscript. We calculated the ratio of coupled electrodes to determine whether there was a difference in the degree of theta-gamma PAC coupling when astronauts performed memory tests on the ground vs. space.	17-19; 542-558; 575-582; 615-617;

Number	Comment	Response	Location
		This result indicated that fewer PACs were detected in the space station than on the ground (Fig. 5-c). We have provided these results in the revised manuscript.	
4	The choice to favor a PAC analysis centered on theta and gamma rhythms limits the neuroscientific scope of this study. In particular, the advantage of having used a system of 62 surface electrodes has been neglected and only some of these electrodes have been chosen according to predetermined theoretical criteria. The major problem which then arises is that throughout the manuscript the authors use scientific elements previously demonstrated as coming from a neuroanatomically and physiologically identified regions of the brain to interpret the signals measured independently by surface electrodes.	We thank the reviewer for the comment to widen the neuroscientific scope of our study. 1. We further calculated all possible PAC pairs ($n = 10$) of the slow-wave phase and fast-wave amplitude between the δ, θ, α, β, and γ frequency range. We found that θ-β, α-β, and α-γ were also significant coupled, but their coupled cortical locations were different from θ-γ (Fig. S6). Moreover, only θ-γ decoupled after the lengthy mental task. We have provided these results with supplementary materials and discussed them in the revised manuscript. 2. Previous studies have suggested that rhythm waves interact through an ensemble of coupling forms in the cerebral cortex. Significantly, Chen et al. (2022, Communications biology) reported that when extending analyses to different cortical areas, a complex network organization of brain-rhythm interactions with pronounced clustering. Thus, we used more electrodes to investigate PAC on a finer scale and clustered them by relative position. Furthermore, these surface electrode clusters were named according to lobe regions of the cerebral cortex, which was in line with peers in rhythmic wave couplings, such as Berger (2019, Nature Communications) and Chen (2022, Communications biology) et al. PAC analysis was performed in each electrode pair between two clusters, and coupled numbers were averaged in a cluster to get a more credible coupling evaluation on cortical area level than previous studies. However, the original manuscript did not explain this purpose clearly, thus creating an ambiguity between neuroanatomically / physiologically identified regions and the signals measured independently by surface electrodes.	172-173; 778-792; S: 57-60;
5	On the contrary, it would have been more advantageous to carry out an analysis of the sources of these activities and not be satisfied with considering that the activities	We thank the reviewer for the comment that an analysis of the sources should be carried out in our study. Unfortunately, due to the limitation of conducting brain	11-13; 38; 123; 130-133;

Number	Comment	Response	Location
	measured by a surface electrode had to correspond to a generator located under this same electrode, as this could lead to abusive interpretations.	research experiments on the space station, we only collected EEG and eye movement data, and high spatial resolution or functional imaging techniques were not applied, even in ground-based experiments. Therefore, for a long time during the revision period, we tried to use the existing data for EEG source analysis. However, we need to accurately identify the phase and amplitude of two rhythmic waves with different frequencies to locate PAC activities. For phases and amplitudes that occur at different frequencies, such as the trough of theta waves that occurs every about 200 ms and the amplitude of gamma waves that occur every 10 ms, it seems impossible to analyze PAC activity with the current proven source analysis methods. Furthermore, if we artificially transform the data to be suitable for EEG source analysis but cannot find evaluation criteria for its localization results, this may create new abusive interpretations. To our knowledge, the current interpretation of the use of EEG signals to study rhythmic coupling remains at the level of cortical regions unless electrodes are implanted into the skull. Of course, for healthy humans, using surface electrodes to study the coupling between cortical regions is more valuable. It will also help reveal how neuronal populations regulate physiological functions. As proposed by Chen et al. Chen (2022, Communications biology), who also used surface EEG signals to analyze brainwave coupling, these empirical findings provide new insights into the basic regulatory mechanisms of diverse physiological states and may guide future efforts to understand the role of microscopic signaling pathways and integration processes in neuronal populations dynamics play in regulating physiological functions. For the problem of abusive interpretations that appeared in our original manuscript, we have added restrictive words 'detected by surface EEG', 'cortical area surface ' or 'reflected in cortical area surface ' to the corresponding parts of the full text. In	143; 145-146; 173; 193-194; 205-206; 236-237; 244; 250; 252-254; 267-268; 270-271; 343-344; 349-350; 360; 381; 383; 390; 406; 415; 443; 446-447; 451; 496; 499; 516; 530; 559-560; 571; 590; 602; 606; 676-678; 696; 745-746; 773-774;

Number	Comment	Response	Location
		addition, we have added a limitation of this study described in the Discussion that all interpretations are based on the cortical areas where the surface EEG signal is detected, not the intracranial source of the signal.	
6	Nowhere (except that eye and blink are removed by ICA) is there any mention of the importance of eye movements and blinks in relation to the oscillations present on the surface of the scalp. It is however obvious that not only at rest condition but also during a work task involving repeated letter reading, eye blinking occurs and is accompanied by an oscillatory activity that can increase the GFP by two to four times for frequencies ranging from 0.5 Hz to 4 Hz and this for an interval ranging from - 250 ms to 750 ms on either side of the blink (Liu et al., 2017). In fact, even if we remove the artefactual component of the blinks, the internal dynamics of the brain producing these events essential to the information processing cannot be neglected. Similarly, microsaccades are capable of producing gamma activity and it is very difficult to identify them, especially in microgravity.	We thank the reviewer for the comment that eye movements and microsaccades should be paid attention to in data processing. In addition, we would like to thank an coauthor of this paper. He has another study on ergonomics that was carried out in parallel with the experiments of this study. To solve this problem proposed by the reviewer, he shared the eye-tracking data collected in sync with the EEG. Recently, Liu et al. (2020) provided compelling new evidence that blink-related processes are dynamically modulated not only to accommodate internal cognitive loading demands but also to address specific sensory processing requirements due to differences in the external environment. It is evident that not only at rest condition but also during a work task involving repeated reading, eye blinking occurs and is accompanied by an oscillatory activity (Liu et al., 2017). In this study, astronauts performed WM tasks involving letter switching in microgravity conditions, where microsaccades are easily caused and capable of producing gamma activity. Spike potentials (SP) have received attention because even involuntary microsaccades ($< 1^\circ$) during attempted fixation generate sizeable SPs, which introduce a broadband artifact in the time-frequency spectrum of the EEG, affecting the low-amplitude beta and gamma bands (> 30 Hz) (Yuval-Greenberg et al., 2008), in particular. Dimigen (Dimigen, 2019) demonstrated that correction could be strongly improved by training the ICA on optimally filtered data in which SPs were massively overweighted. With optimized procedures, ICA removed virtually all artifacts from both viewing paradigms, including the SP and its associated spectral broadband artifact, with little distortion of neural activity. Thus, we employed this advanced ICA algorithm	208; 211; 216; 219; 220; 222; 229-230; 291; 293; 303; 306-309; 311-312; 314; 316-317; 392; 399-401; 404-405; 419; 423-424; 426; 439-440; 447; 458; 460; 466-470; 498-502; 518; 525-558; 986-1016

Number	Comment	Response	Location
		based on SP overweighting. In the first step, four parameters of the ICA pipeline were varied orthogonally: the (1) high-pass and (2) low-pass filter applied to the training data, (3) the proportion of training data containing myogenic saccadic SP, and (4) the threshold for eye tracker based component rejection. In the second step, the eye-tracker was used to objectively quantify the correction quality of each ICA solution, both in terms of under-correction (residual artifacts) and overcorrection (removal of neurogenic activity). Dimigen (2019) provided Matlab code in the published paper. Applying this artifact removal method, we recalculated all EEG data and updated these results in the revised manuscript.	
7	In summary, if the quality of the signals measured in the Chinese space station is comparable to that carried out on the ground, which an analysis of the SNR in the two situations should demonstrate, these data deserve to be more exploited, by researching with broader perspective and more powerful tools for analyzing the specificity of cerebral functioning in weightlessness.	We thank the reviewer for the comment that we should compare the quality of the signals measured in the space station and on the ground. Therefore, occur times per minute of spontaneous eye blinks and saccadic eye movements were counted according to eye-tracking data to analyze the SNR in the two situations. The statistical results show that the frequency of blink and saccade of astronauts in the space station is higher than that on the ground (Fig. S7-a), indicating that the quality of the signals measured in the Chinese space station was worse than that carried out on the ground. So, we employed the above advanced ICA algorithm to remove artifacts in EEG signals. Furthermore, to assess the effectiveness of the ICA-based artifact removal procedure, quantitative analyses were performed to evaluate the changes in regional signal power before and after artifact rejection, referencing Liu et al. (2017). Results showed that the power ratio was significantly reduced in the FCC and TPC regions following artifact removal, and there was no significant difference in the signals measured in the Chinese space station and on the ground (Fig. S7-b and c) after artifact removal. This result indicated that the properties consistent with ocular artifact had been successfully removed from the EEG data. We have provided this artifact removal	986-1016 S: 61-67;

Number	Comment	Response	Location
		method in the revised manuscript and these results with supplementary materials.	

REVIEWERS' COMMENTS:

Reviewer #1 (Remarks to the Author):

The authors have fully addressed my comments and suggestions in the revised manuscript and added supplemental material. I now recommend publication.

Reviewer #2 (Remarks to the Author):

The authors took my remarks into account and their revised manuscript correctly mentions these different elements, which makes the discussion of these original results more relevant.

Referee comments and responses checklist

We replied to all referee comments within the table below. Each reply is presented together with the relevant referee comment. Comments are numbered in one column and duplicated in the second column. Responses are listed in the third column.

Number	Comment	Response
Reviewer #1:		
1	The authors have fully addressed my comments and suggestions in the revised manuscript and added supplemental material. I now recommend publication.	We thank the reviewer for the recommendation of our research.
Reviewer #2:		
1	The authors took my remarks into account and their revised manuscript correctly mentions these different elements, which makes the discussion of these original results more relevant.	We thank the reviewer for commenting on our research content.